# Separable Pathways for Causal Reasoning: How Architectural Scaffolding Enables Hypothesis-Space Restructuring in LLM Agents

## Abstract

Causal discovery through experimentation and intervention is fundamental to robust problem solving. It requires not just updating beliefs within a fixed framework but revising the hypothesis space itself, a capacity current AI agents lack when evidence demands representations they have not previously constructed. We extend the blicket detector paradigm from developmental science to test this capacity in AI agents equipped with architectural scaffolding that targets hypothesis-space restructuring. Our compositional architecture has two discrete components: context graphs, which structure exploration as typed state machines, and dynamic behaviors, which monitor for evidence that the current hypothesis space is inadequate and expand it at runtime. Across 1,085 experimental trials, these components make orthogonal contributions: context graphs drive reasoning quality within the post-switch hypothesis space, accounting for 94% of the accuracy gain, while dynamic behaviors drive reasoning eligibility by detecting regime changes and preventing premature commitment to outdated hypotheses. The benchmark codebase, all agent implementations, trace data, and analysis scripts are publicly available.

## 1 Introduction

Causal reasoning is the capacity to construct, test, and revise explanatory models of the world. This reasoning underpins generalization, efficient exploration, and robust adaptation in ways that associative learning alone cannot enable (Pearl, 2009; Spirtes et al., 2000). Developmental studies show that even young children leverage causal models to learn from sparse evidence (Gopnik et al., 2004; Gopnik & Wellman, 2012), actively constructing interventions to disambiguate competing hypotheses rather than passively accumulating observations (McCormack et al., 2016). For agentic AI systems that must act under uncertainty and learn from their own interventions, this capacity is what distinguishes robust problem-solving from brittle pattern matching (Schölkopf et al., 2021; Lake et al., 2017).

Yet current AI systems exhibit a striking gap in precisely this capability. While LLMs can generate correct causal arguments with high probability (Kıcıman et al., 2024), this performance reflects causal knowledge from training corpora rather than genuine discovery through intervention. Yiu et al. (2024) described this as "cultural transmission": the aggregation and retransmission of human-generated knowledge, fundamentally different from the capacity to discover novel causal structure. Zečević et al. (2023) reached a similar conclusion through so-called meta structural causal models, finding that LLMs reproduce correlations over causal facts encountered during training. Empirical evaluations confirm this pattern: LLMs perform near chance on causal inference tasks that cannot be solved by retrieval (Jin et al., 2024), and their apparent causal competence degrades on questions post-dating training data (Chi et al., 2024); see Section 2.1 for a detailed review. Larger models retrieve more causal facts; they do not acquire the capacity to revise their causal framework when those facts prove inadequate.

What would genuine causal reasoning require? Developmental research offers a precise answer. Children do not simply learn which variables are causal within a fixed rule structure. They learn to revise the rule

structure itself, forming and revising abstract "overhypotheses", or beliefs about the kind of causal rule that governs a domain (Gopnik & Wellman, 2012). Kemp et al. (2007) formalized overhypothesis learning as hierarchical Bayesian inference, showing that learners can acquire abstract structural knowledge (e.g., "this domain uses conjunctive rules") from the same data used for first-order learning (e.g., learning which specific variables are causal). A key result of this and related work is that acquiring the right hypothesis space often accelerates learning more than improved selection within a fixed one (Tenenbaum et al., 2011).

In the hierarchical Bayesian framework, these levels are unified: the same inference process that updates first-order beliefs also updates the higher-order hypothesis space. The compositional architecture evaluated here takes a different approach, factorizing the problem into two discrete components whose contributions are empirically isolable. One component structures inference within a fixed hypothesis space, what we term *reasoning quality*. The other component, *reasoning eligibility*, monitors for evidence that the space itself is inadequate and expands it by adding representational dimensions that the agent's state space did not previously contain. We refer to this expansion as *hypothesis-space restructuring*. This is not a reweighting of hypotheses within a fixed representational vocabulary but an expansion of the vocabulary itself by adding dimensions that the agent's state space did not previously contain. Our central claim is that this factorization is productive: the two components will make orthogonal contributions to distinct failure modes.

We instantiate them as components of a compositional architecture designed so that each layer's contribution is empirically isolable through ablation. Context graphs, drawing on statechart formalisms (Harel, 1987), structure the agent's problem-solving process as an explicit graph of reasoning phases, supporting reasoning quality within a fixed hypothesis space. Dynamic behaviors, rooted in runtime verification (Leucker & Schallhart, 2009), operate as monitors layered on top of the state machine, detecting when the current hypothesis space cannot account for the evidence and triggering state-space reorganization. Together they provide the machinery for hypothesis-space restructuring.

To evaluate this architecture, we extend the blicket detector paradigm, originally developed to study causal learning in children (Gopnik & Sobel, 2000; Gopnik et al., 2004) and formalized for computational agents by Kosoy et al. (2022a). In the standard task, an agent must discover which objects activate a novel machine and what rule governs activation. We introduce conditions that require hypothesis-space restructuring: the generative rule changes mid-experiment and the agent must discover that new causal dimensions are relevant, not simply re-evaluate existing variables.

This article makes three contributions:

- **The Extended Blicket Benchmark.** We extend the Kosoy et al. (2022a) paradigm into a parameterized evaluation suite that tests hypothesis-space restructuring specifically, introducing a novel hidden moderator condition where the causal rule changes mid-experiment. We also contribute a diagnostic metric, *reasoning-eligible accuracy*, that disentangles structural traps from genuine reasoning failures.[1]

- **The Separable Pathways Finding.** We show that the two architectural components make orthogonal contributions: context graphs drive reasoning quality within the post-switch hypothesis space, while dynamic behaviors drive reasoning eligibility by detecting regime changes and preventing premature commitment to outdated hypotheses. Neither substitutes for the other.

- **Boundary Conditions.** The architectural advantage is specific to hypothesis-space restructuring. The benchmark includes two additional extended conditions on which the current scaffolding provides no reliable benefit: order-sensitive (placement sequence is causally relevant) and stochastic (activation is probabilistic). These null results establish where the architecture's coverage ends and serve as diagnostic targets for future enrichment.

---

[1] The benchmark codebase, all agent implementations, trace data, and analysis scripts have been anonymized and included in this submission. Upon acceptance, these artifacts will be made publicly available.

## 2 Related Work

### 2.1 Causal Learning, Overhypotheses, and the Blicket Paradigm

The so-called theory-theory framework established that children construct and revise causal models of their environment, treating learning as a process of theory change rather than associative accumulation (Gopnik et al., 2004; Gopnik & Wellman, 2012). An important experimental design for this work is the blicket detector paradigm (Gopnik & Sobel, 2000). In this paradigm, the detector is a device that activates (lights up, plays music) when certain objects, called blickets, are placed on it according to a hidden causal rule controlled by the experimenter. The participant explores the environment by placing and removing objects, observing activation feedback, and forming hypotheses about which objects are causally active and how. Dependent measures across two decades of blicket studies include object-level categorization ("Is this a blicket?"), novel intervention design ("Make the machine go/stop"), and exploration behavior metrics such as steps to activation and exploration strategy (Gopnik & Sobel, 2000; Gopnik et al., 2001; Sobel et al., 2004; Griffiths et al., 2011; Kosoy et al., 2022a; Jiang & Lucas, 2024).

These studies reveal a hierarchical structure in human causal learning. Children do not merely learn which objects are blickets. They form overhypotheses: transferable abstract beliefs about what kinds of causal rules are possible (Kemp et al., 2007; Goodman, 1955). The belief that the detector responds to object combinations (a conjunctive overhypothesis) operates at a higher level of abstraction than any particular object assignment and greatly constrains the first-order search. Tenenbaum et al. (2011) showed that hierarchical Bayesian models capture this "blessing of abstraction," demonstrating how overhypotheses acquired in one domain transfer to novel ones and accelerate learning. More recent work provides direct evidence that humans deliberately pursue overhypothesis information during active causal learning, selecting interventions that are informative about abstract causal structure rather than the immediate task (Jiang & Lucas, 2024).

On the computational side, the capacities that make human causal learning effective remain elusive for AI agents. These limitations are quantified by recent benchmarking work. Jin et al. (2024) found that seventeen LLMs, including GPT-4, performed near chance on the Corr2Cause benchmark for pure causal inference, with fine-tuned models failing to generalize under distribution shift. Chi et al. (2024) showed that performance drops sharply on causal questions post-dating training data. Evaluations using the blicket paradigm confirm this at the task level. Kosoy et al. (2022b) found that deep RL and behavior cloning agents could learn causal structures seen during training but could not generalize to held-out configurations, while LLMs (GPT-3, PaLM) could not reason about causal structure from observations alone. GX-Chen et al. (2025) showed that LM agents exhibit a systematic disjunctive bias on the blicket task, reliably inferring disjunctive rules but failing on conjunctive ones, an effect mirroring adult human biases rather than the unbiased exploration characteristic of young children. Dhole (2026) included blicket-detector inference in BabyReasoningBench, finding that smaller language models improve with scale but remain far from flexible hypothesis revision. The common thread is that no evaluated system, whether task-specific or general-purpose, demonstrates the capacity to revise its hypothesis space when the current one proves inadequate.

### 2.2 State Machines and Structured Agent Architectures

State machines provide a natural formalism for structuring multi-step agent behavior. Recent work has applied this principle to LLM agent control. Wu et al. (2024) introduced StateFlow, modeling LLM task-solving as state machines with customizable transitions and demonstrating that structured workflows outperform unstructured prompting on multi-step tasks. A common insight is that externalizing the agent's decision-making topology and making it inspectable and controllable, rather than leaving it implicit in the LLM's chain of thought, yields consistent performance gains. However, in all existing approaches, the state topology is determined before the episode begins and cannot change in response to evidence gathered during execution. For example, automated workflow design methods optimize the graph structure but likewise fix it at execution time (Zhang et al., 2025). This is the limitation the present work addresses: when the agent encounters phenomena its current structure cannot represent, the structure itself must be expandable at runtime.

### 2.3 Runtime Behavior Adaptation and Dynamic Behaviors

The problem of adapting agent behavior at runtime, selecting or reorganizing behavioral components in response to changing evidence, predates the LLM era. Behavior trees (Colledanchise & Ögren, 2018) and the options framework (Sutton et al., 1999) both provide mechanisms for choosing among candidate strategies based on current context. In all cases, however, the repertoire of candidate strategies is fixed at design time, and none modify the agent's problem space in response to what those strategies detect. Our dynamic behaviors address this problem; their design and mechanism are described in Section 3.3.

## 3 Methods: Architecture and Ablation Design

### 3.1 The Three-Agent Hierarchy

We evaluate three agents arranged as an additive hierarchy: a base LLM without architectural scaffolding, and two compositional architectures that each contribute a distinct structural function. The hierarchy is compositional in the sense that each successive layer adds a structurally distinct capacity without removing or modifying the one below it. Consequently, performance differences between agents adjacent on the hierarchy are attributable to the added component. This ablation-by-design principle draws on a broader tradition in modular and compositional architectures, where task performance is decomposed by composing specialized modules whose individual contributions can be isolated (Andreas et al., 2016; Yang et al., 2024). The Base agent receives only a task description in its system prompt. The CG agent adds a context graph that structures exploration into typed states with explicit objectives and transition conditions (Section 3.2). The CG+DB agent further adds dynamic behaviors that monitor agent reasoning at runtime and expand the problem space when evidence warrants it (Section 3.3). Table 1 summarizes the architectural properties of each agent. We describe each component below in sufficient detail for replication, and the released codebase supports extension to new domains.

Table 1: Agent architectural properties.

| Property | Base | CG | CG+DB |
|---|---|---|---|
| Task description in system prompt | Yes | Yes | Yes |
| Context Graph (exploration structure) | No | Yes (4 states) | Yes (4 + up to 4 dynamic) |
| State transitions declared by agent | No | Yes | Yes |
| Dynamic Behaviors (runtime adaptation) | No | No | Yes (4 behaviors) |
| System notifications injected mid-episode | No | No | Yes |

All three agents receive an identical task description that specifies the environment (i.e., a blicket detector that activates when certain objects are placed on it), the object set, and the two candidate rule types (conjunctive and disjunctive). The task description does not reference phenomena from the extended conditions, like evidence of a hidden moderator; the agent must discover these through interaction. The full task description is reproduced in Appendix A.1.

All agents use Anthropic's Sonnet 4.5 as the action and reasoning model. The CG+DB agent additionally uses Haiku 4.5 for trigger evaluations within the dynamic behavior pipeline, separating the computational cost of monitoring from the primary reasoning process. Both models were called at temperature 1.0 with no top-k restriction, so that episode-level variability reflects the natural sampling distribution of the underlying models; the random seeds reported in Table 4 govern environment generation only, not model inference.

### 3.2 Context Graphs as Problem Space Definitions

A context graph is a directed graph of typed states. Each state carries a name (e.g., INITIAL_EXPLORATION), a type (one of ACTION, DECISION, or REFLECTION), an objective specifying what the agent should accomplish, ordered guidelines for how to proceed, and transitions to other states

with human-readable conditions. The context graph is injected into the system prompt at every step (Appendix A.2 provides an example rendering). The agent therefore knows its current position in the graph, the objective and guidelines for that state, and the set of legal transitions. To move between states, the agent includes a `TRANSITION: STATE_NAME` directive in its response; transitions are validated against the graph topology so that only structurally permitted moves are accepted. This mechanism gives the agent deliberate control over its exploration strategy while constraining it to the problem space defined by the graph.

The base context graph for the blicket task has four states arranged in a deliberate progression that mirrors how a disciplined investigator would approach the problem: explore, combine, reflect, verify. Back-edges allow the agent to return to a prior state when evidence is ambiguous. Figure 1 (left) illustrates the four-state graph with state names, types, objectives, guidelines, and back-edges; Figure 1 (right) shows the expanded graph after a dynamic behavior detects a rule change (Section 3.3).

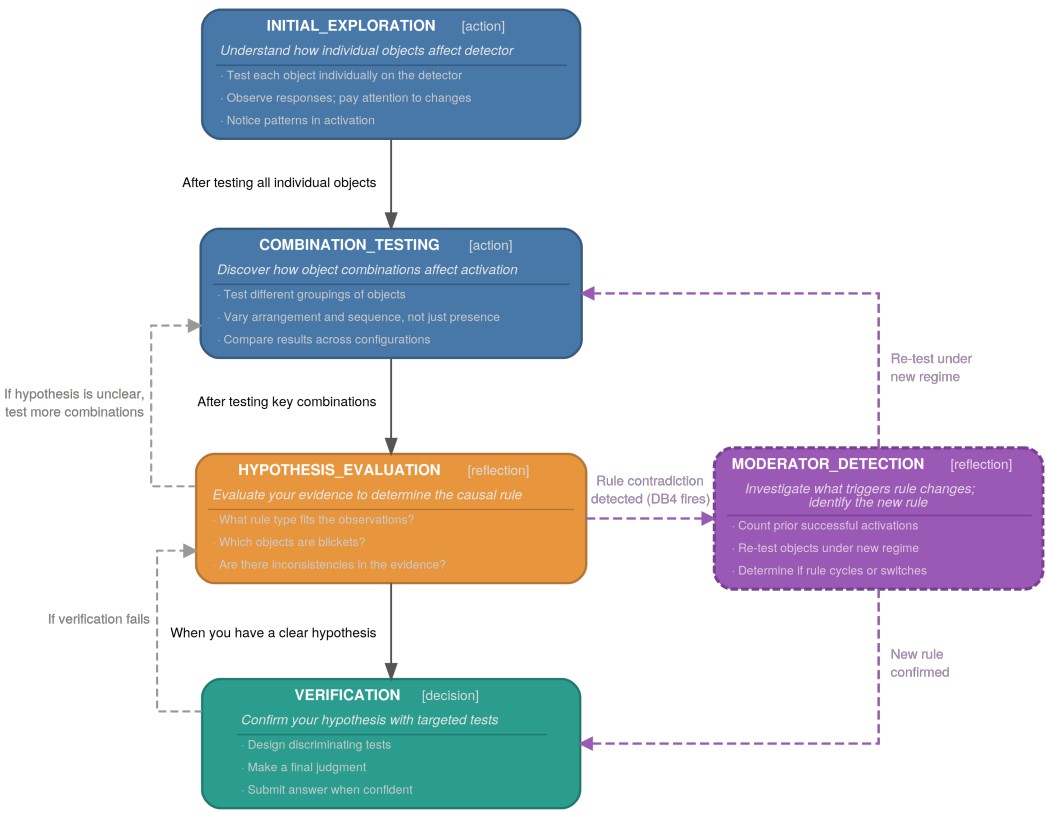

Figure 1: Context graph architecture. Left: the base 4-state graph. Right: the expanded graph after a dynamic behavior (DB4) fires.

The four states and their roles are as follows:

- **INITIAL_EXPLORATION** (action) tests individual objects one at a time. Guidelines direct the agent to observe responses, attend to changes, and notice patterns. The transition condition ("after testing all individual objects") is evaluated by the LLM, not enforced programmatically.

- **COMBINATION_TESTING** (action) tests object groupings. Guidelines explicitly reference varying arrangement and sequence, not just presence or absence.

- **HYPOTHESIS_EVALUATION** (reflection) steps back from acting to evaluate accumulated evidence. Guidelines pose questions about rule type, object identity, and inconsistencies.

- **VERIFICATION** (decision) designs discriminating tests and makes a final judgment. If verification fails, the agent returns to HYPOTHESIS_EVALUATION.

The standard conjunctive and disjunctive rules for activating the blicket detector are part of the vocabulary of these four states. An agent operating within this graph can revise its causal model from, for example, a disjunctive rule with a single blicket to a conjunctive rule with two blickets. The prompts injected at every turn include terms like "individual objects," "combinations," and "rule type," allowing the agent to navigate the standard experimental conditions efficiently. The four-state graph has no vocabulary, however, for phenomena presented in the extended conditions described in Section 4. For example, there is no concept of rule change as distinct from a stable rule, which is required for the hidden moderator condition. Without a mechanism to expand the graph at runtime, the CG agent is confined to reasoning about novel phenomena within states that lack the vocabulary to represent them.

### 3.3 Dynamic Behaviors as Computational Overhypotheses

Dynamic behaviors are runtime monitors that observe agent exploration and fire when evidence suggests a phenomenon the context graph cannot represent. When a behavior fires, it adds new states and transitions to the graph and injects a system notification into the agent's context informing it that the problem space has been expanded. The agent does not request these notifications; they arrive as a consequence of the trigger evaluation pipeline described below. Each behavior fires at most once per episode. This mechanism implements hypothesis-space restructuring.

We propose that dynamic behaviors constitute a computational analogue to overhypotheses (Kemp et al., 2007; Goodman, 1955). The mapping holds across four dimensions:

- **Overhypothesis ↔ dynamic behavior.** An overhypothesis is an abstract belief about what kinds of causal relationships are possible. Each dynamic behavior encodes exactly such a belief: rule_change_hypothesis encodes "the underlying rule might have changed"; order_hypothesis encodes "placement sequence might be causally relevant."

- **Evaluation ↔ Trigger scoring.** Overhypothesis evaluation assesses whether current evidence supports a particular abstract causal structure. Dynamic behavior trigger evaluation scores (0–10) how strongly current observations support the encoded hypothesis, using a lightweight LLM evaluator (Haiku 4.5).

- **Selection ↔ Competitive inhibition.** Overhypothesis selection chooses which abstract framework best explains the data. A competitive inhibition mechanism selects which dynamic behavior fires: when multiple behaviors score above threshold, only the highest-scoring fires, with at most one runner-up permitted within a margin of 1.0 points.

- **Application ↔ CG modification.** Overhypothesis application constrains the learner's hypothesis space. Dynamic behavior firing modifies the context graph by adding states with new objectives and guidelines, expanding the agent's hypothesis space based on which overhypothesis is selected.

Table 2 summarizes the four dynamic behaviors and their corresponding overhypotheses. Full specifications are provided in Appendix A.3.

We illustrate the full mechanism using DB4 (rule_change_hypothesis), the behavior relevant to the hidden moderator condition (Figure 1, right). The trigger pipeline proceeds in three stages. First, a deterministic pre-screen checks whether minimum conditions are met: at least 10 steps must have elapsed, and the behavior must not have already fired. Unlike DB1, which requires that the detector has never activated, DB4 has no stagnation requirement; it targets situations where a previously reliable rule has broken down. Second, if the pre-screen passes, a condensed context (the last 5 conversation messages, the current context graph

Table 2: Dynamic Behaviors and overhypothesis mappings.

| Dynamic Behavior | Overhypothesis | New CG State | Condition |
|---|---|---|---|
| DB1: exploration_stagnation | "Standard rules are insufficient" | DIMENSION_DISCOVERY | All extended |
| DB2: order_hypothesis | "Placement sequence is causally relevant" | ORDER_TESTING | Order-sensitive |
| DB3: stochasticity_hypothesis | "Activation is probabilistic" | RELIABILITY_TESTING | Stochastic |
| DB4: rule_change_hypothesis | "The causal rule has changed" | MODERATOR_DETECTION | Hidden moderator |

state, and an exploration summary) is sent to Haiku 4.5 with a behavior-specific evaluation prompt: "Has the agent found a rule that worked consistently for several trials but then stopped working?" Haiku returns a score from 0 to 10. If the score meets or exceeds the threshold (6.0), the behavior fires. Third, upon firing, DB4 adds a MODERATOR_DETECTION state (reflection) to the context graph. This state carries the objective "The causal rule appears to have changed. Investigate what triggers rule changes and identify the new rule," along with guidelines directing the agent to count prior successful activations, re-test objects under the new regime, and determine whether the rule cycles or switches permanently. New transitions connect MODERATOR_DETECTION from HYPOTHESIS_EVALUATION and to COMBINATION_TESTING and VERIFICATION. A system notification is injected into the conversation informing the agent that the new state is available.

Neither component of this architecture alone produces hypothesis-space restructuring. A context graph without dynamic behaviors defines a fixed hypothesis space. While it can support revision between conjunctive and disjunctive hypotheses, it cannot expand to accommodate phenomena outside its vocabulary. Dynamic behaviors without a context graph can detect anomalies but have no structured space to modify. Together, they implement a factorized analogue of the hierarchical learning structure that Tenenbaum et al. (2011) established. Where hierarchical Bayesian models unify first-order and structural inference in a single process, the compositional architecture separates them into discrete components: learning specific causal relationships within context graph states, and evaluating abstract structural principles via dynamic behavior monitoring that can expand the space of representable relationships.[2]

### 3.4 Trace Structure and Analytical Affordances

Each episode produces a structured JSON trace that records the complete agent-environment interaction: every action taken, every environment response, the context graph state at each step, all dynamic behavior trigger evaluations and firings, and the agent's final answer. These traces are the raw data from which all reported results are derived. The full trace schema and an annotated example are provided in the released codebase.

The trace structure supports analysis along several dimensions. Accuracy analysis decomposes errors into rule-type confusion (conjunctive vs. disjunctive) and object-set errors (over-inclusion or under-inclusion of candidate blickets). Step efficiency measures how many of the budgeted actions the agent consumed before submitting an answer. Context graph state progression reveals the agent's exploration strategy, including the sequence and timing of state visits and transitions into dynamically added states like MODERATOR_DETECTION. Dynamic behavior activation logs record which behaviors fired, at what step, and

---

[2]A critic of our ablation design might object that a static context graph could in principle include states for every conceivable phenomenon (e.g., regime changes, stochastic activation, order effects), eliminating the need for dynamic behaviors entirely. However, dynamic behaviors preserve a parsimonious four-state graph representation and expand it only when observations warrant revision, matching the developmental evidence that overhypotheses are recruited by evidential pressure rather than maintained as a standing inventory. The factorization also makes the empirical contribution of each component isolable, which a monolithic graph would not.

with what evidence, enabling mechanistic attribution of performance differences to specific detection events. Transition logs capture both successful and failed state transitions, where failed attempts indicate the agent sought states not yet available in the graph. For the hidden moderator condition, pre- and post-switch behavior can be compared directly to determine whether the agent detected the rule change and adapted its strategy. When episodes fail, the trace supports fine-grained failure classification: wrong rule type, wrong objects, premature termination, or step-budget exhaustion.

The trace structure is shared across all three agent types, which makes cross-architecture comparison possible at the turn level, not just at the episode-outcome level. This granularity is what enables the reasoning eligibility analysis introduced in Section 5.

## 4 Methods: The Extended Blicket Benchmark

### 4.1 The Blicket Detector Environment

Panel 1 summarizes the environment; we describe the interaction cycle here. On each turn, the agent selects an action, placing or removing an object, and observes whether the detector is ACTIVE or INACTIVE. Consider a conjunctive episode with blickets A, B among five objects. The agent places A alone: INACTIVE. Places B alone: INACTIVE. Places A and B together: ACTIVE. Removes A, leaving only B: INACTIVE. From these four observations, the agent has evidence that neither A nor B is individually sufficient but both together are required. The remaining objects C, D, and E must also be tested to confirm they are distractors, and the agent must rule out the possibility that a larger set (e.g., A, B, C) is required.

---

**Panel 1: The Blicket Detector Environment**

| | |
|---|---|
| **Objects** | A set of named objects: A, B, C (3-object variant) or A, B, C, D, E (5-object variant). Some are *blickets* (causally active); the rest are distractors. |
| **Detector** | A device that is either ACTIVE (glowing) or INACTIVE. Activation is governed by a hidden causal rule that determines when the detector activates based on which objects are present. |
| **Hidden rule** | The causal rule is not revealed to the agent. It must be inferred from the pattern of activations and non-activations observed during exploration. |
| **Actions** | **Place:** put an object on the detector. **Remove:** take an object off. **Check:** submit a final answer. |
| **Goal** | Identify (1) which objects are blickets, and (2) whether the rule is *conjunctive* (all blickets must be present) or *disjunctive* (any single blicket suffices). |
| **Answer format** | `RULE_TYPE: conjunctive / disjunctive · BLICKETS: A, B` |
| **Step budget** | 50 actions (3-object) or 75 actions (5-object). |
| **Standard conditions** | *Conjunctive:* all blickets must be present simultaneously. *Disjunctive:* any single blicket suffices. These serve as ceiling baselines (100% accuracy, all agents, all runs). |

---

The agent may submit its answer at any point by selecting the Check action, but the step budget is binding. Extended conditions with five objects have a step budget of 75, which is enough for disciplined exploration but not for exhaustive enumeration of all possible object combinations. Agents that submit prematurely

risk missing critical evidence; agents that explore without converging risk exhausting the budget without submitting an answer at all.

## 4.2 Experimental Conditions

The standard conditions, conjunctive and disjunctive, establish whether the agent can reason about causality at all. For example, the agent must understand the logic of conjunction to place two objects together to activate the detector. As all agents perform at ceiling on these (see Panel 1), we introduce three extended conditions that each require the agent to revise a different dimension of its causal framework. The hidden moderator condition is the core experimental condition for this paper; order-sensitive and stochastic serve as boundary conditions that probe the limits of architectural support. Object-to-role assignments (which objects are blickets, distractors, or moderators) are randomized across runs; the examples below use one representative assignment.

**Hidden moderator (core experimental condition).** The rule begins as conjunctive (e.g., A, B) and switches after N successful activations to disjunctive (e.g., C). The agent first establishes that A+B activates the detector, building confidence in a conjunctive model. Then, silently, the rule changes. On the next test, A+B produces INACTIVE, contradicting a model the agent had confirmed multiple times. Testing individual objects reveals that C now activates the detector alone. The agent must abandon its validated model and re-learn the rule from scratch, revising the overhypothesis "there is one stable rule" to "the rule can change based on a hidden condition." This is the most demanding revision: it requires not just detecting an anomaly but recognizing that the entire causal regime has shifted.

**Order-sensitive (boundary condition).** The rule is conjunctive with an order constraint, as in: A, B placed in sequence [A, then B]. Placing A then B activates the detector; placing B then A does not, despite the same objects being present. The agent must discover that placement order is a causally relevant variable, a dimension absent from the conjunctive/disjunctive framework. The hypothesis-space violation occurs when the agent places the same pair that previously worked and gets a different result, with order as the only difference. This condition probes whether the agent can revise the overhypothesis "rules depend only on which objects are present" to include object sequencing.

**Stochastic (boundary condition).** The rule is conjunctive with probabilistic activation (p = 0.70), for example, conjunctive A, B at 70%. The correct objects activate the detector most of the time, but not always. The agent must discover that activation is probabilistic rather than deterministic. This requires revising the overhypothesis "the same configuration always gives the same result." The 70% rate is high enough to be detectable but low enough to create genuine ambiguity: is A+B the wrong combination, or is the rule noisy? Resolving this requires repeated testing and statistical reasoning, a form of inference that proves difficult for all three architectures, making this condition an informative boundary.

Mid-task rule changes such as those probed with the hidden moderator condition have a long history in cognitive assessment, notably the Wisconsin Card Sorting Test (Berg, 1948) and probabilistic reversal learning paradigms. They have recently been applied to LLM evaluation as well (Li et al., 2025). To our knowledge, however, the hidden moderator condition is the first to embed a mid-experiment regime change within a causal learning task where the agent must not only detect that the rule has changed but also identify the blicket objects and rule type from scratch. This is structurally richer than set-shifting between pre-given categories (as in the WCST) or reversing reward contingencies (as in reversal learning): the agent faces an open-ended causal induction problem under a regime it did not know could exist. The structure also has natural real-world analogues, as in a treatment protocol that stops working when a patient's condition changes (see Section 6.4).

The hidden moderator condition is the most demanding test of hypothesis-space restructuring in the benchmark: the agent must detect that the causal regime has shifted and re-learn the rule from scratch, using whatever representational resources its architecture provides.

### 4.3   Metrics

### 4.3.1   Standard Metrics

Each episode yields a set of standard metrics computed directly from the trace. Overall accuracy records whether the agent correctly identified both the rule type (e.g., conjunctive vs. disjunctive) and the exact object set; when incorrect, the error is decomposed into rule-type errors and object-set errors (further classified as over-inclusion or under-inclusion). This measure is strictly harder than the object-level categorization ("Is this a blicket?") used in the developmental literature (Gopnik & Sobel, 2000; Sobel et al., 2004), because an agent that correctly identifies all blickets but infers the wrong rule type, or vice versa, receives no credit. Requiring explicit rule identification forces discrimination at the overhypothesis level, not just the object level. Steps taken measures how many of the budgeted actions the agent consumed before submitting, and answer rate tracks whether the agent submitted an answer at all versus exhausting the step budget. For the hidden moderator condition, switch rate records the proportion of episodes where the agent accumulated enough activations to trigger the rule change; a prerequisite for any post-switch analysis. We also compute exploration metrics including unique configurations tested and steps to first activation, though these serve primarily as diagnostic tools for understanding agent behavior rather than as primary dependent variables. Parse failure rate (steps where agent output could not be mapped to a valid action) is tracked but is negligible across all runs reported here. Dynamic behavior firing patterns (i.e., which behaviors fired, at what step, and with what trigger score) provide the mechanistic complement to these outcome metrics.

### 4.3.2   Reasoning Eligibility and Reasoning-Eligible Accuracy

The hidden moderator condition creates a structural trap that must be addressed analytically. Because the rule switches at an unannounced point, every episode can be classified into one of three categories shown in Table 3 based on the agent's activation count relative to the switch threshold N.

Table 3: Three-way episode classification in the Hidden Moderator condition.

| Category | Condition | What it means |
|---|---|---|
| Pre-switch | activations < N | Rule never changed. Agent answered before reaching the threshold. Accuracy is 100% — trivially correct. |
| Exactly-N | activations == N | Rule switched on the agent's final activation. Agent triggered the switch but submitted before observing any post-switch evidence. |
| Reasoning-eligible (RE) | activations > N | Rule switched AND agent observed at least one post-switch evidence point before submitting. |

The exactly-N failure mode occurs when the agent accumulates exactly N activations, which are enough to trigger the rule switch, but not enough to observe any post-switch evidence. The agent submits a pre-switch answer that is guaranteed to be wrong: a structural trap unique to this condition, not a failure of reasoning. Since accuracy in this category is 0% regardless of architecture, and accuracy in the pre-switch category is 100% with all agents, raw accuracy conflates predetermined outcomes with genuine reasoning differences.

To isolate the effect of architecture on causal reasoning, we define reasoning-eligible (RE) episodes as post-switch episodes where the agent accumulated more than N activations, meaning it continued experimenting after the rule changed and has in principle encountered evidence of the regime change. RE accuracy is then computed as the proportion of correct episodes among RE episodes only. This metric removes the two categories where the outcome is predetermined (pre-switch: always correct; exactly-N: always wrong) and isolates reasoning capacity from structural luck. RE accuracy is the primary dependent variable for all formal hypothesis tests reported in Section 5. This metric was not specified a priori. The exactly-N

failure mode was first identified in Run 04, and the three-way classification was subsequently adopted as the primary analytical framework and applied uniformly across all runs.

## 4.4 Experimental Design Summary

Table 4 lists the runs reported in this paper. Runs 03–08 use the 5-object configuration with three agents (Base, CG, CG+DB). Run 03 is the initial 5-object pilot across all conditions; Runs 04–06 progressively refine the hidden moderator design, culminating in Run 06's pre-registered exploration-depth manipulation; Runs 07–08 introduce harder post-switch rules to break the RE accuracy ceiling and isolate the separable-pathways finding. The stochastic and order-sensitive results from Run 03 serve as boundary conditions. Run 10g is the powered order-sensitive comparison, using a 4-object configuration and testing only Base and CG+DB (the broader 10x series focused on these two architectural endpoints; see Section 5.5). The full experimental series (Runs 01–11e) is documented in the project repo; Runs 01–02 used a 3-object configuration and are excluded from formal analyses due to ceiling effects and an environment bug in the order-sensitive condition. The run series reflects iterative refinement of both the benchmark conditions and the architectural components; the implications of this development process for generalizability are discussed in Section 7.

Table 4: Experimental runs reported in this paper.

| Run | Episodes | Condition | Key Manipulation | Headline Result |
|-----|----------|-----------|------------------|-----------------|
| 03 | 300 | All 4 conditions (5-obj) | 5-object scaling, competitive inhibition | Order at ceiling; stochastic flat (73–77%); hidden mod 93.3% CG+DB ($p \approx 0.08$) |
| 04 | 90 | Hidden moderator (5-obj) | Pre-registered replication ($n = 30$/cell) | Replication confirmed; exactly-N failure mode discovered |
| 05 | 75 | Hidden moderator (5-obj) | Switch-point calibration (sw = 5,7,9) | Base has soft exploration ceiling 4 activations |
| 06 | 300 | Hidden moderator (5-obj) | Exploration depth × architecture (sw = 3,5) | CG+DB > CG ($p$ = 0.015, BF_1_0=100.4); RC at 100% sensitivity |
| 07 | 120 | Hidden moderator (5-obj) | Conjunctive post-switch rules (CD, AC, CDE) | CDE breaks RE ceiling: CG+DB 100% vs Base 80% |
| 08 | 200 | Hidden moderator (5-obj, CDE) | Powered 3-way comparison + runway compression | CG drives 94% of RE gain; DB drives exactly-N avoidance; separable pathways ($p = 0.013$) |
| 10g | 60 | Order-sensitive (4-obj, 100 steps) | Powered 2-agent comparison (Base, CG+DB) | No reliable advantage (+10pp, $p$ = 0.276) |

All statistical analyses are computed from raw trace files via a single reproducible script (available in the project repository) that includes automated regression checks: pre-computed reference values for Run 08 are verified against recomputed values at runtime, and the script terminates if any discrepancy exceeds a tolerance of 0.2%.

All pairwise accuracy comparisons use Fisher's exact test (two-sided for all new analyses; one-tailed where noted for comparisons replicated from earlier run reports, reflecting directional predictions derived from the architectural hierarchy). Effect sizes are reported as Cohen's $h$ (computed as $2(\arcsin \sqrt{p_1} - \arcsin \sqrt{p_2})$) for proportion comparisons, and odds ratios with Wald-type 95% confidence intervals (log OR $\pm$ 1.96 $\times$ SE; Haldane–Anscombe correction applied when any cell count is zero) for the exactly-N analysis. The Cochran–Mantel–Haenszel test is used in Section 5.1 to assess the CG vs. Base comparison stratified by batch (Runs 03, 04, and 06), with continuity correction and Robins–Breslow–Greenland confidence intervals for the common odds ratio. Bayes factors (BF_1_0) are computed as directional Bayes factors via Monte Carlo sampling ($10^6$ draws) from Beta(k+1, $n - k$+1) posteriors under uniform Beta(1,1) priors. BF_1_0 values in Section 5.1 (e.g., BF_1_0=100.4) are reported as posterior odds, P(p_1 > p_2 | data) / P(p_1 ≤ p_2 |

data); supplementary analyses use the convention BF\_1\_0 = P(p\_1 > p\_2 | data) / 0.5. Under equal priors the two are related by BF\_odds = 2P/(1−P) vs. BF\_prob = P/0.5; both draw on identical posterior samples. All statistical computations are implemented in `paper1_statistical_tests.py` (Python; `scipy.stats`, `scipy.special`, `numpy`).

All experiments were run using the Anthropic API. The primary experimental series (Runs 03–08; 1,085 episodes) cost approximately \$807 in total API fees; the full programme spanning 23 runs and 2,541 episodes cost approximately \$1,950. Per-episode costs ranged from \$0.17 (3-object, 20-step budget) to \$3.16 (5-object, 100-step budget with full scaffolding). The CG+DB agent incurs a 30–50% cost premium over Base at matched step budgets, driven primarily by the Haiku-based trigger evaluations at each reasoning step.

## 5 Results

### 5.1 Regime Change Detection Under Easy Post-Switch Rules (Runs 03–06)

The first four runs use an easy disjunctive post-switch rule. In this environment, all agents approach ceiling on reasoning-eligible accuracy, making raw accuracy differences attributable to structural trap exposure rather than reasoning quality. Across three independent batches totaling 110 episodes per agent (Runs 03, 04, and 06; all at switch point $N = 3$), CG+DB achieved 89.1% raw accuracy compared to 77.3% for CG (Fisher's exact $p = 0.015$, one-tailed; BF\_1\_0=100.4). Table 5a presents the combined data; Table 5b provides the batch-level detail.[3]

Table 5: Hidden moderator results with easy (disjunctive) post-switch rule.

(a) Combined sw $= 3$ (Runs 03 + 04 + 06, $n = 110$ per agent)

| Agent | Raw Accuracy | RE Episode Rate | RE Accuracy | Exactly-N Rate |
|---|---|---|---|---|
| Base | 96/110 (87.3%) | 81/110 (73.6%) | 79/81 (97.5%) | 12/110 (10.9%) |
| CG | 85/110 (77.3%) | 75/110 (68.2%) | 74/75 (98.7%) | 24/110 (21.8%) |
| CG+DB | 98/110 (89.1%) | 96/110 (87.3%) | 94/96 (97.9%) | 10/110 (9.1%) |

(b) Batch-level detail (sw $= 3$)

| Batch | Agent | n | Raw Accuracy | RE Episode Rate | RE Accuracy | Exactly-N Rate |
|---|---|---|---|---|---|---|
| Run 03 | Base | 30 | 23/30 (76.7%) | 19/30 (63.3%) | 19/19 (100%) | 7/30 (23.3%) |
| Run 03 | CG | 30 | 23/30 (76.7%) | 21/30 (70.0%) | 21/21 (100%) | 7/30 (23.3%) |
| Run 03 | CG+DB | 30 | 28/30 (93.3%) | 28/30 (93.3%) | 28/28 (100%) | 2/30 (6.7%) |
| Run 04 | Base | 30 | 28/30 (93.3%) | 24/30 (80.0%) | 24/24 (100%) | 2/30 (6.7%) |
| Run 04 | CG | 30 | 21/30 (70.0%) | 20/30 (66.7%) | 20/20 (100%) | 9/30 (30.0%) |
| Run 04 | CG+DB | 30 | 26/30 (86.7%) | 23/30 (76.7%) | 23/23 (100%) | 4/30 (13.3%) |
| Run 06 | Base | 50 | 45/50 (90.0%) | 38/50 (76.0%) | 36/38 (94.7%) | 3/50 (6.0%) |
| Run 06 | CG | 50 | 41/50 (82.0%) | 34/50 (68.0%) | 33/34 (97.1%) | 8/50 (16.0%) |
| Run 06 | CG+DB | 50 | 44/50 (88.0%) | 45/50 (90.0%) | 43/45 (95.6%) | 4/50 (8.0%) |

The CG+DB advantage over CG replicates across all three batches: +16.6pp (Run 03), +16.7pp (Run 04), and +6.0pp (Run 06). The narrower gap in Run 06 is consistent with regression to the mean at the larger sample size. Base raw accuracy, by contrast, shows high batch variance (76.7%, 93.3%, 90.0%), driven by stochastic fluctuation in exactly-N episode counts rather than reasoning differences (Table 5b, Exactly-N Rate column).

---

[3]Run 04 is a pre-registered replication of Run 03 with an independent random seed (seed=251 vs. seed=137). It uses the same configuration (5-object, sw $= 3$, $n = 30$/cell) and its primary analysis pools with Run 03 to form the $n = 60$/cell dataset; Run 06 ($n = 50$/cell) completes the combined $n = 110$/cell analysis.

A counterintuitive pattern emerges in Table 5a: CG performs numerically *worse* than Base on raw accuracy (77.3% vs. 87.3%). This difference is consistent in direction across all three batches but does not reach significance (Fisher's $p = 0.077$; Cochran–Mantel–Haenszel stratified $p = 0.078$, common OR=2.01 [0.98, 4.12]). The source is not reasoning quality (RE accuracy is roughly equivalent, 98.7% vs. 97.5%), but differential exposure to the exactly-N structural trap (21.8% vs. 10.9%). This pattern resolves fully in Section 5.4, where the separable pathways decomposition shows that CG-driven exploration increases trap exposure while simultaneously improving post-switch reasoning.

Run 06 also tested $sw = 5$ ($n = 50$/agent); at this higher switch point, most agents submit before triggering the rule change, and the condition becomes a selective filter on exploration depth rather than reasoning quality. Full $sw = 5$ results are reported in Appendix B.4.

## 5.2 The Rule Change Detection Mechanism

Now that we have established that dynamic behaviors improve raw accuracy through exactly-N avoidance, we examine whether the detection mechanism itself is reliable. DB4 (rule_change_hypothesis) fires when the agent's recent observations contradict its established causal model. Across five independent datasets, DB4 achieved 100% sensitivity: every post-switch episode in which the CG+DB agent successfully adapted to the new rule was preceded by a DB4 firing ($N = 136$ firings; per-run breakdown in Appendix B, Table B2).[4] These datasets span the 3-object pilot (Run 02, excluded from formal analyses but included here as additional evidence for the mechanism) and the 5-object experiments reported in this paper (Runs 03, 04, and 06 at both switch points). Positive predictive value (PPV) ranged from 90.5% to 100% across datasets. The four PPV failures reflect downstream reasoning errors: the mechanism correctly detected the regime change, but the agent misidentified the new rule or exhausted its step budget.

The converse is equally informative: no post-switch episode in the CG+DB agent resulted in successful adaptation without DB4 firing. Sensitivity of 100% establishes DB4 firing as a necessary condition for adaptation; PPV of 90.5–100% establishes it as a nearly sufficient one. This combination makes the detection event not merely correlated with successful adaptation but mechanistically linked to it. DB4 firing triggers a context graph expansion to MODERATOR_DETECTION, which provides structured guidance for re-exploration under the new causal regime (Section 3.3). The chain from trigger (anomalous observation contradicting the established model) to mechanism (DB4 firing at an identifiable step) to architectural response (context graph state addition and transition) to behavioral outcome (systematic re-testing and correct rule identification) is fully observable in the episode trace.

## 5.3 Breaking the Reasoning-Eligible Accuracy Ceiling (Runs 07–08)

Section 5.1 established that RE accuracy under the easy disjunctive post-switch rule is at ceiling for all three agents (97.5–98.7%). The architecture determines whether the agent reaches the reasoning-eligible zone, but not what it does there. To discriminate reasoning quality, we replace the easy post-switch rule with a conjunctive triple C,D,E (one of 10 possible triples from five objects), requiring systematic combinatorial search to identify. Run 07 ($n = 120$) confirmed that this manipulation breaks the ceiling; Run 08 ($n = 50$/agent, 75-step budget) is the powered three-way comparison reported here.

Figure 2 decomposes all 150 Run 08 episodes into four categories: pre-switch correct (trivially correct, no reasoning tested), exactly-N failures (structural trap, 0% accuracy by definition), RE correct, and RE wrong. The visual pattern encodes the separable-pathways finding directly.

The context graph drives reasoning quality. Base RE accuracy is 73.3% (22/30); CG achieves 93.9% (31/33), a gain of +20.6pp (Fisher's $p = 0.038$, Cohen's $h = 0.59$). This step accounts for 94% of the total Base-to-CG+DB improvement. Adding dynamic behaviors yields 95.3% (41/43), a further +1.4pp that is not significant (Fisher's $p = 1.000$). The context graph contribution is visible in Figure 2 as the near-identical RE-correct segments for CG and CG+DB, both substantially taller than Base's. Structured combination

---

[4]Sensitivity is computed over post-switch correct episodes only. An earlier internal analysis reported 94.4% sensitivity for Run 04 using a denominator that included pre-switch episodes (where DB4 correctly does not fire); when properly scoped, sensitivity is 100% across all five datasets.

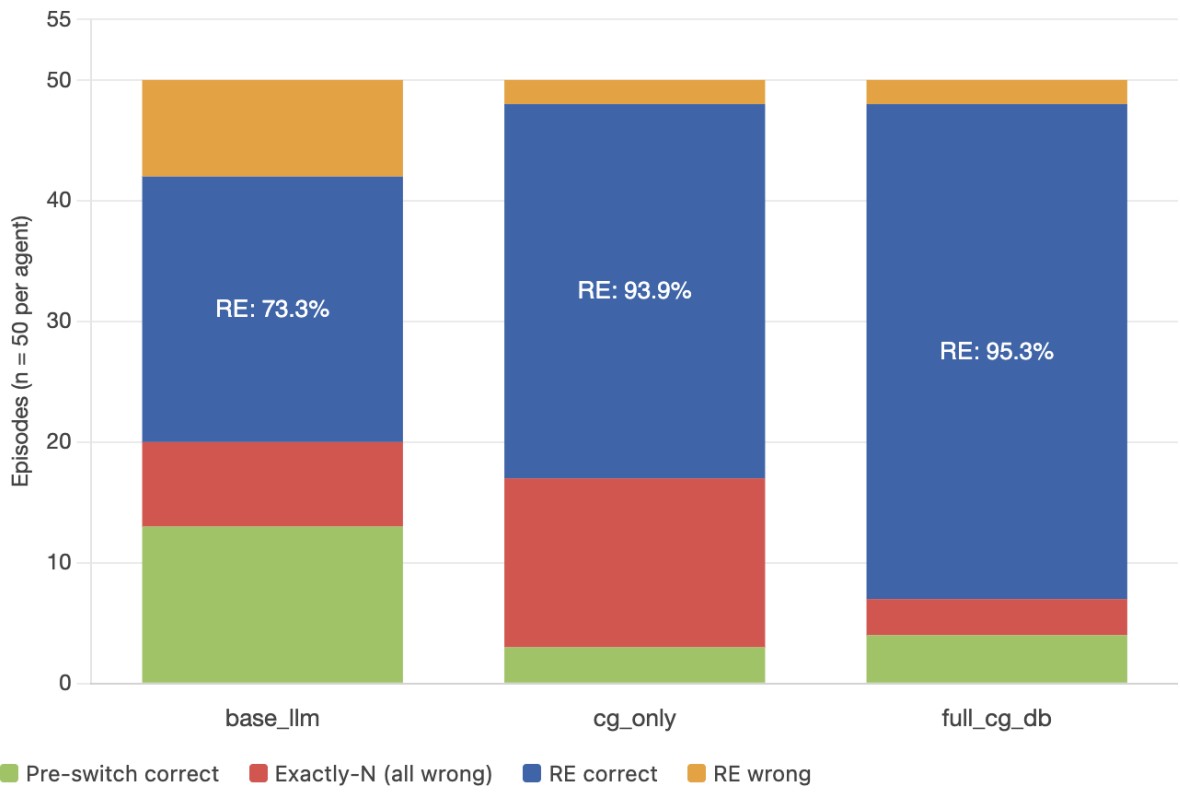

Figure 2: Run 08 episode decomposition by agent ($n = 50$ per agent, 75-step budget, conjunctive post-switch rule {C,D,E}). Each bar shows the four-way classification of episodes. RE accuracy labels are displayed within the RE-correct segment.

testing enables the scaffolded agents to isolate C,D,E from 10 candidate triples; undirected LLM search does not.

Dynamic behaviors drive exactly-N avoidance. The CG exactly-N rate is 28.0% (14/50), compared to 6.0% (3/50) for CG+DB (Fisher's $p = 0.003$, OR=6.09 [1.63, 22.82], Cohen's $h = 0.62$). The Base-to-CG+DB comparison (14.0% vs. 6.0%) is directional but not significant ($p = 0.159$). The asymmetry is visible in Figure 2: CG's exactly-N segment (14 episodes, 28% of its bar) is nearly five times larger than CG+DB's (3 episodes, 6%). The mechanism is specific: context graph-driven exploration reliably triggers the rule switch (94% switch rate vs. 74% for Base), but without DB4's system notification, CG submits its pre-switch hypothesis immediately after the switch, before re-exploring.

Error analysis confirms the decomposition. All 33 "A,B" errors across all three agents occur in exactly-N episodes, an exceptionless mapping to the structural trap. No reasoning-eligible episode in any cell submitted the pre-switch answer. Base RE errors are uniformly over-inclusion: all 8 incorrect RE episodes at 75 steps guessed all five objects A,B,C,D,E, indicating that the unscaffolded agent detects the rule change but cannot isolate the triple through undirected search. Scaffolded-agent RE errors are rare (4 total across CG and CG+DB at 75 steps; 6 across both step budgets) and heterogeneous. Full error tables are provided in Appendix B.

Runway compression provides a robustness check. Reducing the step budget from 75 to 50 does not degrade RE accuracy for either Base (73.3% vs. 80.0%, $p = 0.726$) or CG+DB (95.3% vs. 90.0%, $p = 0.586$). The raw accuracy drop for CG+DB at 50 steps ($90.0\% \rightarrow 72.0\%$) is driven by an increased exactly-N rate (16.0% vs. 6.0%), not by degraded reasoning. The compressed runway gives DB4 less time to fire before the agent submits. Full runway analysis is given in Appendix B.

### 5.4 The Separable Pathways Finding

The results from Sections 5.1–5.3 converge on a single structural claim: the compositional architecture's two components address failure modes that are not merely different but orthogonal, as summarized in Table 6.

Table 6: Separable pathways summary (Run 08, conjunctive post-switch rule C,D,E).

| Pathway | Component | Mechanism | Key evidence |
|---|---|---|---|
| Reasoning quality | Context Graph | Structured combination testing under hard post-switch rule | RE accuracy: Base 73.3% $\rightarrow$ CG 93.9% (+20.6pp, $p = 0.038$, Cohen's $h = 0.59$) |
| Reasoning eligibility | Dynamic Behaviors | System notification prevents premature submission after rule switch | Exactly-N rate: CG 28.0% $\rightarrow$ CG+DB 6.0% ($p = 0.003$, OR=6.09, Cohen's $h = 0.62$) |

Adding dynamic behaviors to a context graph yields no significant RE accuracy improvement (93.9% $\rightarrow$ 95.3%, $p = 1.000$), confirming that dynamic behavior operates on eligibility, not quality. The context graph accounts for 94% of the total 22.0pp RE accuracy gain from Base to CG+DB; dynamic behavior accounts for 6%.

The apparent context graph harm in raw accuracy reported in Section 5.1 (Base 87.3% vs. CG 77.3%, $p = 0.077$) resolves entirely under this decomposition. When exactly-N episodes are excluded, CG significantly outperforms the bare LLM on reasoning-eligible accuracy (93.9% vs. 73.3%, $p = 0.038$). The source of the raw-accuracy harm is not reasoning but structural trap exposure: context graph-driven exploration is systematic enough to reach the switch threshold in 94% of episodes (vs. 74% for Base), but without dynamic behavior's revision cue, the agent submits before re-exploring. The context graph's liability is exactly-N exposure, not reasoning quality. The dynamic behavior's contribution is trap prevention, not reasoning improvement.

The starkest illustration of pathway independence is CG's aggregate performance. Despite achieving 93.9% accuracy when reasoning is possible, significantly better than the bare LLM, its raw accuracy (68.0%) is the lowest of all three agents, because the CG-driven exploration that enables superior reasoning also maximizes structural trap exposure. Together, context graphs and dynamic behaviors achieve what neither can alone: CG+DB reaches 95.3% reasoning-eligible accuracy and 6.0% exactly-N rate, yielding 90.0% raw accuracy (Figure 2). The compositional architecture requires both components precisely because they address orthogonal failure modes.

Our orthogonality claim is functional, not architectural. Dynamic behaviors require a context graph to modify, and so the two mechanisms are coupled at the implementation level. However, they address statistically separable failure modes, as evidenced by the fact that each component moves one metric without significantly affecting the other.

### 5.5 Boundary Conditions: Specificity of the Architectural Advantage

Two additional conditions probe whether the architectural advantage generalizes beyond hypothesis-space restructuring. Table 7 summarizes results across all four conditions.

On the stochastic condition, where the detector activates probabilistically ($p = 0.70$), all three agents converge to 73–77% accuracy (Table 7). We know that the architecture detects the phenomenon because the stochasticity dynamic behavior fires in 77% of stochastic episodes. But detection does not translate to improved performance. The downstream context graph response lacks structured mechanisms for evidence accumulation and statistical decision-making. The agent correctly diagnoses that evidence may be unreliable but still cannot distinguish a relevant object that failed to activate ($p = 0.70$) from an irrelevant object that never activates.

Table 7: Boundary conditions. Hidden moderator rows reproduce the primary findings from Sections 5.1 and 5.3 for comparison. Stochastic results are from Run 03 (5-object, $n = 30$/agent). Order-sensitive results are from Run 10g (4-object, 100-step budget, $n = 30$ pooled); the 10x series tested only Base and CG+DB.

| Condition | Base | CG | CG+DB | Advantage? | Interpretation |
|---|---|---|---|---|---|
| Hidden moderator (easy, raw acc) | 87.3% | 77.3% | 89.1% | Yes ($p = 0.015$) | Hypothesis-space restructuring |
| Hidden moderator (hard, RE acc) | 73.3% | 93.9% | 95.3% | Yes ($p = 0.038$) | Reasoning quality under revision |
| Stochastic | 76.7% | 76.7% | 73.3% | No | Statistical inference (flat) |
| Order-sensitive | 20% | — | 30% | No ($p = 0.276$) | Information-sparse environments |

The order-sensitive condition, where the agent must discover that placement sequence is causally relevant, showed no reliable architectural advantage across multiple runs in the 10x series (Runs 10–10g). The powered comparison (Run 10g: 4-object, 100-step budget, $n = 30$ pooled, excluding the CG agent) yielded 30% for CG+DB vs. 20% for Base ($p = 0.276$). The condition operates in a binary regime: trivially solved at low object counts, intractable at higher counts without richer activation feedback. It appears there is no stable discrimination zone between architectures.

These boundary conditions strengthen the hidden moderator finding. If the architecture improved performance uniformly, as a general scaffolding effect, gains on stochastic reasoning and order discovery would be expected as well. Instead, the advantage is specific to tasks requiring hypothesis-space restructuring, where the causal dimensions have changed and the hypothesis space must be expanded. It is absent for tasks requiring statistical inference within a fixed framework (stochastic) or combinatorial search in information-sparse environments (order-sensitive).

## 6 Discussion

### 6.1 Hypothesis-Space Restructuring as an Architectural Capability

We argue that the gap between human and AI causal learning is partially architectural, and not purely a function of training data or scale. In hierarchical Bayesian models, first-order inference and structural revision are unified in a single process (Kemp et al., 2007). The compositional architecture evaluated here factorizes them into discrete components. Agents learn specific causal relationships within context graph states, for instance by systematically testing object combinations to isolate a conjunctive triple, while dynamic behavior monitoring evaluates abstract structural principles and can expand the hypothesis space itself. The separable pathways finding (Section 5.4) confirms that this factorization is functionally meaningful: the two components make orthogonal contributions to distinct failure modes. This factorization trades away the bidirectional coupling of hierarchical Bayesian models, where first-order evidence continuously modulates structural beliefs and vice versa. In the compositional architecture, the interaction is unidirectional: dynamic behaviors expand the hypothesis space, but within-state reasoning does not gradually shift over-hypothesis strength. The tradeoff is deliberate: what factorization loses in inferential integration it gains in empirical isolability and auditability, as demonstrated by the separable pathways decomposition in Section 5.4 and the mechanistic attribution given in Section 5.2. To our knowledge, these results provide the first evidence that architectural mechanisms, rather than scaling, can enable hypothesis-space restructuring in AI agents on a task designed to test this capability.

In the terminology of Yiu et al. (2024), scale broadens the repertoire of retrievable causal knowledge but does not produce causal innovation. The separable pathways finding (Section 5.4) makes this concrete.

Context graphs account for 94% of the total RE accuracy gain (Section 5.4);[5] dynamic behaviors are the sole mechanism for exactly-N trap avoidance, reducing the structural trap rate from 28.0% to 6.0% ($p = 0.003$). These are not the same mechanism scaled up. They are orthogonal mechanisms addressing orthogonal failure modes: within-regime inference and between-regime detection, respectively. The overhypothesis analogy (Section 3.3) captures the design rationale for dynamic behaviors: each encodes an abstract causal belief and evaluates it against evidence. However, the measured contribution on this benchmark is concentrated in exploration continuation (exactly-N avoidance) rather than reasoning quality per se. Whether dynamic behaviors can improve RE accuracy under conditions where the base agent's reasoning is weaker remains an open question for harder benchmark configurations.

Our compositional architecture also connects to the sparse mechanism shift hypothesis (Schölkopf et al., 2021), which holds that real-world distributional changes are typically localized: one or a few causal mechanisms shift while the rest remain stable. The hidden moderator condition instantiates this structure directly. The activation rule changes, from conjunctive A, B to disjunctive C, but the environment mechanics, object set, action space, and the existence of a learnable rule all remain intact. The architectural decomposition mirrors this sparse structure: context graphs handle inference within the stable post-switch regime, while dynamic behaviors handle detecting that a regime change has occurred. The separable pathways finding is, in this framing, an empirical reflection of the alignment between problem structure and architectural decomposition. If real-world causal changes tend to be sparse, as Schölkopf et al. argue, then an architecture that decomposes shift detection from within-regime inference has structural reasons to generalize beyond the empirical performance observed on a synthetic benchmark like ours.

## 6.2 Interpretability

While all three agents generate complete trace data, the architectural scaffolding provides a distinct auditability benefit. In the base LLM, successful adaptation to a regime change is distributed across the conversation: the agent may gradually drift toward the correct answer, or arrive at it for reasons difficult to reconstruct from the chain-of-thought alone. There is no single discrete event identifiable as the moment of framework revision. In the CG+DB agent, DB4 firing is exactly such an event. It is a timestamped entry recording that the rule-change monitor evaluated the agent's recent experience, scored the evidence above threshold, added a new state to the context graph, and injected a system notification. The 100% sensitivity and 97.1% PPV reported in Section 5.2 establish this firing as a necessary and nearly sufficient condition for successful post-switch adaptation, not merely a correlate. This level of mechanistic attribution, linking a specific detection event at a specific step to a specific behavioral outcome, is unusual in agent evaluation, where scaffolding typically improves aggregate performance without indicating why any individual episode succeeded or failed. Our claim is not that the architecture makes an opaque system transparent; the base LLM adapts successfully in 73.3% of RE episodes without any such mechanism. Rather, it makes the critical reasoning event discrete, locatable, and auditable. This distinction matters less in research, where an analyst can read both traces, than in deployment contexts where audit trails are a requirement and interpretable models should replace post-hoc explanations (Rudin, 2019). For example, a clinical healtcare agent that reports "at step 34, the rule-change monitor detected a contradiction with confidence 8.2 and triggered a protocol reassessment" provides a qualitatively different basis for oversight than a reconstructed chain-of-thought. This interpretability operates at the architectural level and is complementary to, not a substitute for, mechanistic interpretability of the underlying model (Bereska & Gavves, 2024).

## 6.3 The Extended Blicket Benchmark as a Research Instrument

The Extended Blicket Benchmark is a contribution independent of the compositional architecture evaluated on it. We introduced a new experimental condition, the hidden moderator, to evaluate hypothesis-space restructuring, and in the process uncovered two methodological contributions. The exactly-N failure mode revealed that a substantial fraction of errors on regime-change tasks are structural traps: episodes where the agent never encounters post-switch evidence and therefore cannot reason about the new rule regardless

---

[5]GX-Chen et al. (2025) showed that bare LLMs inherit adult-like disjunctive biases from training data, a limitation that scaling reproduces rather than resolves. Our CG's structured combination testing closes a 20pp gap on precisely the conjunctive rules where this bias bites hardest, demonstrating architectural scaffolding overcoming what scale cannot.

of capability. The three-way episode classification this enabled (i.e., pre-switch, exactly-N, and reasoning-eligible) disentangles structural confounds from genuine reasoning failures and is reusable for any evaluation involving hidden state changes. Burnell et al. (2023) argued that aggregate evaluation metrics routinely obscure critical performance variation across instance subgroups. The exactly-N discovery is a clear instance of this problem: raw accuracy aggregates over episodes that are structurally unsolvable, masking the true capability picture. Reasoning-eligible accuracy is the granular decomposition they advocate.

The benchmark's parameterized design also exposes a difficulty hierarchy that resists the saturation problem plaguing many static benchmarks. Under standard conditions, all agents perform at ceiling (Section 5.1), as do all agents on the easy disjunctive post-switch rule under RE accuracy (Section 5.3). Only the hard conjunctive triple C,D,E discriminates reasoning quality across the architectural hierarchy (Sections 5.3–5.4). Researchers can use the benchmark to dial up complexity, for example with more objects, harder post-switch rules, higher switch points, and use these parameterizations to stay ahead of rapidly improving models. Furthermore, the boundary conditions we established for rule order and stochastic rule application are diagnostics, not failures. The null effects for the stochastic condition (Section 5.5) indicate that hypothesis-space restructuring and statistical inference under noise are distinct capabilities. The order-sensitive condition's instability (Section 5.5) suggests that some task dimensions require richer feedback mechanisms, such as partial activation signals or graded confidence, to create a stable discrimination zone. Both point to concrete directions for the next generation of benchmark extensions.

Finally, the hidden moderator condition is deliberately synthetic, but its structure has direct analogues in applied settings where established decision rules silently become invalid, as in treatment protocols that stop working when resistance patterns shift, diagnostic rules that break in new patient populations, or similar structures in financial regulation, cybersecurity, and manufacturing quality control. The parameterized design of the benchmark (variable switch points, post-switch rule complexity, and object counts; see Appendix A.4) supports rapid extension to domain-specific instantiations where object identities map to real variables and rule types map to domain categories. The benchmark is released as a research instrument, and we hope it proves useful to the community well beyond the architectural questions explored here.

### 6.4 Future Work

In our current setup, agent memory resets between episodes, so our experiments test *selection* among pre-compiled overhypotheses rather than *induction* of new ones. A stronger form of agentic causal reasoning would involve cross-episode induction: an agent that, after encountering hidden moderators in episodes 1–N, develops an expectation for rule changes in episode N+1 with entirely novel objects. This expectation derives not from a designer-specified overhypothesis, but because the agent induced it from its own interaction history. The current experiments demonstrate that the architecture can restructure its hypothesis space when evidence warrants it, but the set of possible restructurings is fixed at design time. Cross-episode induction would test whether the repertoire of overhypotheses itself can expand from experience, shifting the locus of causal learning from the designer to the agent.

The Extended Blicket Benchmark supports a direct test. A multi-episode protocol with novel object labels each time would ensure that only abstract structural knowledge can transfer, mirroring the cross-instance overhypothesis transfer documented in human learners (Lucas & Griffiths, 2010; Lucas et al., 2014). The primary prediction is that a memory-augmented agent detects rule changes earlier and solves post-switch rules faster in later episodes, with a learning curve that stabilizes as the induced overhypothesis consolidates. If an architecture can support not only selection among overhypotheses but their induction from experience, it would narrow the gap between AI agents and the hierarchical learning documented in humans (Tenenbaum et al., 2011), demonstrating again that the gap is addressable by compositional structure, not scaling alone.

## 7 Limitations

**Single model family.** All experiments use Claude Sonnet 4.5 as the action model and Haiku 4.5 for trigger evaluation. The architectural contribution is model-agnostic in principle: context graphs and dynamic behaviors impose no model-specific assumptions. But the empirical results are demonstrated on one model

family. Cross-model replication (e.g., GPT-4, Gemini, open-weight models) would strengthen the generalizability claim. This limitation also opens a fruitful question: does the architecture help more when the base model is weaker? The composition approach taken here predicts yes, and testing with a smaller action model would address both generalizability and this theoretical prediction simultaneously.

**Synthetic environment.** The blicket detector is structurally equivalent to the paradigm introduced by Kosoy et al. (2022a) and inherits its experimental validity for studying causal learning. However, it is not ecologically valid for clinical, industrial, or other applied settings. The current results should be interpreted as demonstrating an architectural capacity, not a deployment-ready capability. Section 6.3 discusses domain-adapted variants as a concrete next step. Additionally, the dynamic behaviors are human-authored rather than generated by the agent, which contrasts with the developmental context where children construct their own overhypotheses. Section 6.4 discusses cross-episode induction as a path toward closing this gap.

**Sample sizes.** The primary result (CG+DB vs. CG raw accuracy, BF_1_0=100.4) reflects a large effect detectable at the sample sizes used. Smaller effects, particularly the Base-to-CG RE accuracy comparison under the easy post-switch rule (where all agents approach ceiling), may require substantially larger samples to resolve. The pooled design ($n = 110$ per agent across three independent batches) mitigates single-run variance but does not eliminate it.

**Proprietary models.** The use of closed-source models limits exact reproducibility of specific numerical results. The benchmark codebase, trace format, and analysis scripts are fully released, and the architectural patterns reported here (separable pathways, exactly-N structure) are testable on any model.

**One-shot dynamic behavior firing.** Each dynamic behavior fires at most once per episode in the experimental configuration. Alternative deployments could allow re-firing with decay, which might alter the exactly-N avoidance rate. The one-shot constraint is in a sense conservative: it establishes that a single detection event is sufficient for adaptation but does not test whether repeated monitoring would improve performance on episodes where the first firing occurs late.

**Iterative development.** The experimental series spans 11+ runs during which both the benchmark conditions and the architectural components (context graph state design, dynamic behavior trigger logic, and competitive inhibition parameters) were iteratively refined. This creates a risk that the architecture is overfit to the specific benchmark rather than exhibiting a generalizable capacity. Two features of the results mitigate this concern without eliminating it. First, the primary effect sizes are large (Cohen's $h = 0.62$ for exactly-N avoidance, h=0.59 for RE accuracy gain, BF_1_0=100.4 for the pooled raw accuracy comparison) and replicate across independent batches with different random seeds. This is a pattern more consistent with a genuine architectural match than with tuning artifacts. Second, the boundary conditions provide a built-in control: the same iterative process that refined the hidden moderator condition also attempted to improve performance on stochastic and order-sensitive conditions, including extensive context graph redesigns (four major revisions across the Run 11 series) and dynamic behavior reconfigurations. These efforts produced no reliable gains, suggesting that the hidden moderator advantage reflects a specific alignment between architecture and task structure rather than general overfitting to the benchmark. Cross-benchmark validation on tasks with analogous regime-change structure would be the definitive test.

## 8 Conclusion

This article makes three contributions. The Extended Blicket Benchmark introduces mid-experiment causal regime changes to the blicket detector paradigm, providing a parameterized evaluation suite for hypothesis-space restructuring with an accompanying methodological refinement, reasoning-eligible accuracy, that disentangles structural traps from genuine reasoning failures. The separable pathways finding demonstrates that the two architectural components make orthogonal contributions: context graphs drive reasoning quality within the post-switch hypothesis space (accounting for 94% of the RE accuracy gain), while dynamic behaviors drive reasoning eligibility by detecting regime changes and preventing premature commitment to outdated hypotheses. The boundary conditions confirm that this advantage is specific to tasks requiring hypothesis-space restructuring and absent for statistical inference under noise or combinatorial search in information-sparse environments. The core claim is that hypothesis-space restructuring is an architectural

capability that emerges from the interaction of structured state representations and runtime monitoring, and is not reducible to either component alone.

The benchmark is released as a research instrument. Its parameterized design supports extension to new models, new agent architectures, and new domains, with variable switch points, post-switch rule complexity, object counts, and step budgets as adjustable parameters. The hidden moderator structure has natural analogues wherever trusted protocols lose validity without signaling the change, and healthcare-adapted variants are a concrete next step. Cross-episode induction, testing whether successful revision in one episode accelerates revision in the next, and cross-model replication are the immediate priorities for future work. Two decades of developmental research have shown that children revise their causal frameworks through the interaction of structured prior knowledge and active exploration. The compositional architecture tested here implements a computational analogue of that interaction. The gap between AI and children on causal learning may be narrower than current evaluations suggest. This is not because AI systems are more capable than previously thought, but because the right architectural scaffolding can unlock capabilities that scaling alone does not provide.

### 8.1 Broader Impact Statement

This work introduces a benchmark and architectural evaluation for causal reasoning in AI agents. The benchmark environment is synthetic, and no claims are made about deployment readiness. Discussion of applied analogues (e.g., clinical protocols, financial regulation) serves to illustrate the structural relevance of hypothesis-space restructuring, not to advocate for autonomous deployment. Any application of agents that autonomously revise their decision frameworks in safety-critical domains would require human oversight, domain-specific validation, and regulatory review beyond the scope of this research.

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

# A Context Engineering Prompts

This appendix reproduces the prompts and structured instructions provided to the agents. The base_llm agent receives only the task description (A.1). The cg_only agent additionally receives the context graph rendering (A.2). The full_cg_db agent receives both, plus dynamic behavior notifications (A.3) and trigger evaluation prompts (A.4).

### A.1 Task Description

All three agents receive the following task description in their system prompt (shown here for the 5-object variant used in Runs 03–08):

```
You are interacting with a 'blicket detector' --- a device
that glows when certain objects (called 'blickets') are
placed on it.

There are 5 objects: A, B, C, D, and E. Some may be
blickets, some may not.

The detector follows a hidden rule:
- CONJUNCTIVE: ALL blicket objects must be on the detector
  for it to activate.
- DISJUNCTIVE: ANY single blicket object is enough to
  activate the detector.

Your goal: Figure out which objects are blickets AND whether
the rule is conjunctive or disjunctive.

You can place objects, remove objects, and observe whether
the detector activates. When you're confident, submit your
answer.

Respond with:
REASONING: <your current thinking>
ACTION: <number or action name>
```

The task description is identical across all conditions. For extended conditions (order-sensitive, stochastic, hidden moderator), the agent receives no advance indication that the causal rule may involve phenomena beyond the conjunctive/disjunctive distinction.

### A.2 Context Graph Rendering

The `cg_only` and `full_cg_db` agents receive a context graph rendering appended to their system prompt each turn. The rendering includes three layers: the current state's objective and guidelines, the available transitions, and a condensed graph overview. The following is an example rendering with the agent in COMBINATION_TESTING:

```
=== Context Graph ===
Current State: COMBINATION_TESTING [action]
Objective: Discover how object combinations and arrangements affect activation.
Guidelines:
  - Test different groupings of objects on the detector
  - Vary not just which objects are present, but how you arrange
    and sequence them
  - Compare results across different configurations
  - If a configuration that should work doesn't, consider what
    else might matter
Transitions:
  -> HYPOTHESIS_EVALUATION: After testing key combinations
  -> INITIAL_EXPLORATION: If results are confusing, go back to singles
```

```
Graph overview:
  [action] INITIAL_EXPLORATION -> COMBINATION_TESTING
  [action] COMBINATION_TESTING -> HYPOTHESIS_EVALUATION,
           INITIAL_EXPLORATION <-- YOU ARE HERE
  [reflection] HYPOTHESIS_EVALUATION -> VERIFICATION,
               COMBINATION_TESTING
  [decision] VERIFICATION -> HYPOTHESIS_EVALUATION
```

To change states, the agent includes `TRANSITION: STATE_NAME` in its response. Transitions are validated against the graph topology; only edges present in the graph are accepted. The rendering provides three layers of information: (1) current state details, including the objective, guidelines, and available transitions; (2) a condensed graph overview with a positional marker; and (3) for the full_cg_db agent only, a modification log showing the last three runtime changes when dynamic behaviors have fired. The full state definitions for the base 4-state graph are specified in the codebase.

## A.3 Dynamic Behavior Specifications

This section provides the full specification of each dynamic behavior, including trigger conditions, evaluation prompts, context graph modifications, connectivity, and the system notification injected when the behavior fires. All trigger evaluations use the following standardized prompt template, sent to Haiku 4.5 with a condensed context (last 5 conversation messages truncated to 300 characters each, the current context graph state, and an exploration summary). Haiku returns a score from 0 to 10; behaviors fire when the score meets or exceeds the configured threshold.

```
You are evaluating whether a specific condition has been met
in an agent's exploration of a causal learning task.

Condition to evaluate:
{behavior-specific evaluation prompt}

Current CG state: {current_state}
Exploration summary: {summary}
Recent conversation:
{last 5 messages, 300 chars each}

Rate how strongly this condition is met on a scale of 0-10,
where 0 = definitely not met and 10 = clearly met.
Respond with just the number.
```

Each behavior fires at most once per episode.

### A.3.1 DB1: exploration_stagnation

**Overhypothesis:** Standard conjunctive/disjunctive rules are insufficient.

**Trigger conditions:** stagnation_mode (detector has never activated), min_steps = 6, stagnation_threshold = 6, fire_threshold = 6.0.

**Evaluation prompt:** "The agent has been exploring the detector for several steps but has NEVER achieved activation. Are they stuck in an ineffective exploration pattern? Consider whether they need to think about alternative causal mechanisms beyond simple conjunctive/disjunctive rules."

**CG modification:** Adds DIMENSION_DISCOVERY (reflection). Objective: consider alternative causal dimensions including order, probabilistic activation, and rule changes. Connected from HYPOTHESIS_EVALUATION and COMBINATION_TESTING; transitions to COMBINATION_TESTING and VERIFICATION.

**Notification:** "[SYSTEM] Your observations don't fit standard causal rules. A new exploration state DIMENSION_DISCOVERY is now available. Consider transitioning there to explore alternative causal dimensions."

### A.3.2 DB2: order_hypothesis

**Overhypothesis:** Placement sequence is causally relevant.

**Trigger conditions:** min_steps = 8, fire_threshold = 6.0. No stagnation requirement.

**Evaluation prompt:** "Has the agent observed inconsistent or surprising results that could be explained by the ORDER or SEQUENCE of object placement mattering? For example: the same set of objects producing different results, or configurations that 'should' work based on object identity alone failing to activate."

**CG modification:** Adds ORDER_TESTING (action). Objective: systematically vary placement order while keeping the same objects. Connected from DIMENSION_DISCOVERY and COMBINATION_TESTING; transitions to HYPOTHESIS_EVALUATION and VERIFICATION.

**Notification:** "[SYSTEM] Evidence suggests placement order may matter. A new ORDER_TESTING state is available for systematic order experiments."

### A.3.3 DB3: stochasticity_hypothesis

**Overhypothesis:** Activation is probabilistic rather than deterministic.

**Trigger conditions:** min_steps = 8, fire_threshold = 6.0.

**Evaluation prompt:** "Has the agent tried the EXACT same configuration of objects MORE THAN ONCE and gotten DIFFERENT results? This would suggest probabilistic/stochastic activation rather than a deterministic rule."

**CG modification:** Adds RELIABILITY_TESTING (action). Objective: repeat the same configuration multiple times and estimate activation probability. Connected from DIMENSION_DISCOVERY and COMBINATION_TESTING; transitions to HYPOTHESIS_EVALUATION.

**Notification:** "[SYSTEM] Inconsistent results detected for same configurations. A new RELIABILITY_TESTING state is available to measure activation probability."

### A.3.4 DB4: rule_change_hypothesis

**Overhypothesis:** The underlying causal rule has changed.

**Trigger conditions:** min_steps = 10, fire_threshold = 6.0.

**Evaluation prompt:** "Has the agent found a rule that WORKED CONSISTENTLY for several trials but then STOPPED working? The same objects that reliably activated the detector no longer do so, suggesting the underlying rule has changed."

**CG modification:** Adds MODERATOR_DETECTION (reflection). Objective: investigate what triggers rule changes and identify the new rule. Guidelines direct the agent to count prior successful activations, re-test all objects individually under the new regime, re-test combinations, and consider whether the rule cycles or switches permanently. Connected from DIMENSION_DISCOVERY and HYPOTHESIS_EVALUATION; transitions to COMBINATION_TESTING and VERIFICATION.

**Notification:** "[SYSTEM] The causal rule appears to have changed. A new MODERATOR_DETECTION state is available to investigate rule shifts."

## A.4 Experimental Parameters and Design Rationale

Key parameters reflect choices informed by the iterative run series (Runs 1–11). This section documents values and reasoning to support interpretation and replication.

**Object count (5).** The standard blicket paradigm uses 3 objects (Gopnik & Sobel, 2000; Kosoy et al., 2022a). Pilot runs revealed ceiling effects at 3 objects: all agents solved standard conditions trivially. Scaling to 5 objects (2 blickets, 3 distractors) expanded the combinatorial space sufficiently to differentiate agents on extended conditions while preserving solvability.

**Step budget (75).** Standard conditions use 50 steps; extended conditions use 75. Runs 10e–10f tested 75, 100, and 125 steps on order-sensitive. At 125 steps, DB-driven premature convergence reversed the architectural advantage. The 75-step budget ensures the constraint is binding without truncating viable strategies.

**DB fire thresholds (6.0).** All four behaviors use a uniform threshold of 6.0. Run 2 analysis suggested higher thresholds (7.0+) could reduce false-positive fires, but uniform values were retained to avoid condition-specific tuning that would complicate interpretation.

**One-shot DB firing.** Each behavior fires at most once per episode. The production architecture supports repeatable firing with decay, but one-shot firing isolates the effect of a single revision event. This is noted as a limitation in Section 7.

**Trigger evaluation model.** Trigger evaluations use a lighter model than the agent's reasoning model, separating monitoring cost from primary reasoning and ensuring the two pathways operate at different capability levels — consistent with the claim that monitoring and reasoning are separable functions.

**Switch point ($N$).** The hidden moderator rule change occurs after the agent accumulates $N$ successful activations. This is an experimenter-specified parameter; the agent receives no signal that a switch has occurred. Lower values (e.g., $N = 3$) ensure most agents encounter the post-switch regime; higher values (e.g., $N = 5$) act as selective filters on exploration depth.

## B Supplementary Results

### B.1 Runway Compression (Run 08)

Run 08 tested three agents at 75 steps ($n = 50$/agent) and two agents at 50 steps ($n = 25$/agent) on the conjunctive post-switch rule C,D,E. Table B1 compares RE accuracy and exactly-N rates across step budgets.

Table 8: Runway compression comparison (75-step vs. 50-step).

| Agent | Metric | 75-step | 50-step | Fisher's p |
|-------|--------|---------|---------|------------|
| Base | RE accuracy | 73.3% (22/30) | 80.0% (12/15) | 0.726 |
| Base | Raw accuracy | 70.0% (35/50) | 68.0% (17/25) | — |
| CG+DB | RE accuracy | 95.3% (41/43) | 90.0% (18/20) | 0.586 |
| CG+DB | Raw accuracy | 90.0% (45/50) | 72.0% (18/25) | — |
| CG+DB | Exactly-N rate | 6.0% (3/50) | 16.0% (4/25) | — |

Neither agent shows significant RE accuracy degradation at 50 steps. The raw accuracy drop for CG+DB ($90.0\% \to 72.0\%$) is driven by the increased exactly-N rate ($6.0\% \to 16.0\%$) under the compressed runway, not by degraded reasoning quality.

### B.2 DB4 Rule-Change Detection Performance by Run

Table B2 reports the sensitivity and positive predictive value of DB4 (rule_change_hypothesis) across all runs where CG+DB faced the hidden moderator condition. Sensitivity is computed over post-switch correct episodes only (see Section 5.2, footnote 4).

Run 02 uses the 3-object configuration excluded from formal analyses (Section 4.4) but is included here as additional evidence for the mechanism. All four false positives reflect episodes where DB4 correctly detected

Table 9: DB4 detection performance (CG+DB only).

| Run | Config | Switched | RC Firings | TP | FP | FN | Sensitivity | PPV |
|---|---|---|---|---|---|---|---|---|
| 02 | 3-obj, sw = 3 | 24 | 20 | 19 | 1 | 0 | 100% | 95.0% |
| 03 | 5-obj, sw = 3 | 30 | 28 | 28 | 0 | 0 | 100% | 100% |
| 04 | 5-obj, sw = 3 | 27 | 23 | 23 | 0 | 0 | 100% | 100% |
| 06 (sw = 3) | 5-obj, sw = 3 | 49 | 44 | 43 | 1 | 0 | 100% | 97.7% |
| 06 (sw = 5) | 5-obj, sw = 5 | 27 | 21 | 19 | 2 | 0 | 100% | 90.5% |
| **Total** | | **157** | **136** | **132** | **4** | **0** | **100%** | **97.1%** |

the regime change but the agent failed downstream (misidentified the new rule or exhausted the step budget). False negatives are zero across all datasets: no post-switch episode was solved without DB4 firing.

### B.3 Error Classification (Run 08)

Table B3 classifies all errors from the 75-step cells of Run 08.

Table 10: Error summary by agent (75-step cells, $n = 50$/agent).

| Agent | Total Errors | Exactly-N | RE Errors | RE Error Detail |
|---|---|---|---|---|
| Base | 15 | 7 | 8 | 8× over-inclusion (A,B,C,D,E) |
| CG | 16 | 14 | 2 | 1× over-inclusion, 1× timeout |
| CG+DB | 5 | 3 | 2 | 1× over-inclusion, 1× superset (A,C,D,E) |
| **Total** | **36** | **24** | **12** | |

All 33 exactly-N errors across all five Run 08 cells (24 from the 75-step cells in Table B3 above, plus 9 from the two 50-step cells not shown) submitted the pre-switch answer A,B without exception. No reasoning-eligible episode in any cell submitted A,B. The mapping between the structural trap and the A,B answer pattern is exceptionless.

Base RE errors are uniformly over-inclusion: all 11 RE errors across both step budgets (8 at 75 steps, 3 at 50 steps) guessed all five objects A,B,C,D,E. The agent detects the rule change but cannot isolate the triple through undirected search. Scaffolded-agent RE errors are rare (6 total across both agents and both step budgets) and heterogeneous: 3 over-inclusions, 1 superset, 1 timeout, and 1 wrong rule type (disjunctive, in the 50-step CG+DB cell).

### B.4 Switch-Point 5 Results (Run 06)

Run 06 tested sw = 5 ($n = 50$/agent) alongside the sw = 3 condition reported in Section 5.1. At sw = 5, the switch point is high enough that most agents submit before triggering the rule change, and the condition becomes a selective filter on exploration depth. Table B4 reports the full results.

Table 11: Hidden moderator results at sw = 5 (Run 06, $n = 50$ per agent, easy disjunctive post-switch rule).

| Agent | Raw Accuracy | RE Episode Rate | RE Accuracy | Exactly-N Rate |
|---|---|---|---|---|
| Base | 50/50 (100.0%) | 3/50 (6.0%) | 3/3 (100%) | 0/50 (0.0%) |
| CG | 45/50 (90.0%) | 4/50 (8.0%) | 4/4 (100%) | 5/50 (10.0%) |
| CG+DB | 42/50 (84.0%) | 19/50 (38.0%) | 19/19 (100%) | 8/50 (16.0%) |

RE episode rates diverge sharply: CG+DB reaches the reasoning-eligible zone in 38% of episodes, compared to 8% for CG and 6% for Base. Among the few episodes that do reach the zone, RE accuracy is 100% for all three agents. The easy disjunctive post-switch rule is trivially solvable once the agent has any post-switch evidence; the architecture's contribution at this switch point is purely to exploration depth, not reasoning quality.

