# OpenReview forum: "Separable Pathways for Causal Reasoning: How Architectural Scaffolding Enables Hypothesis-Space Restructuring in LLM Agents"
_TMLR — Under review for TMLR_

### Review · Reviewer_ufSt · 2026-05-08

**Summary Of Contributions:**

The paper investigates whether LLM agents can revise their hypothesis space during causal reasoning when existing assumptions fail. It extends the blicket detector task with mid-experiment rule changes and evaluates a compositional agent architecture combining context graphs for structured exploration with dynamic behaviors for detecting regime shifts and expanding the agent’s reasoning states at runtime. Through comparisons among a base LLM, a context-graph agent, and a full context-graph-plus-dynamic-behavior agent, the paper shows that structured exploration improves post-switch reasoning quality, while dynamic behaviors help agents avoid premature commitment to outdated hypotheses.

**Audience:**

Yes

**Audience Explanation:**

The paper is relevant to TMLR because it studies the design and behavior of LLM-based agents in a causal-reasoning task. It proposes an extended benchmark, analyzes agent behavior through ablations, and provides findings about how structured state representations and runtime monitoring affect causal adaptation. This should interest readers working on agent architectures, causal reasoning, and cognitively motivated evaluation.

**Broader Impact Concerns:**

The paper already includes a Broader Impact Statement, and I do not see major ethical concerns beyond those discussed.

The main concern is possible over-interpretation of the results for high-stakes domains such as healthcare, finance, or regulation. The paper should continue to emphasize that the benchmark is synthetic and does not demonstrate deployment readiness.

If the authors keep real-world analogies, they should explicitly state that autonomous agents that revise decision frameworks in safety-critical settings require human oversight, domain-specific validation, and regulatory review.

**Claims And Evidence:**

Yes

**Claims Explanation:**

The submission’s main claims are mostly supported by the reported evidence:

1. **LLM agents struggle with causal reasoning when the rule structure changes.**
   This is tested through the hidden moderator condition, where the causal rule changes mid-experiment and the agent must revise its earlier hypothesis.

2. **Context graphs improve post-switch reasoning quality.**
   This is supported by the Run 08 ablation, where reasoning-eligible accuracy improves from 73.3% for Base to 93.9% for CG.

3. **Dynamic behaviors reduce premature commitment to outdated hypotheses.**
   This is supported by the drop in exactly-N failures from 28.0% for CG to 6.0% for CG+DB.

4. **The two components play separable roles.**
   The evidence is that CG mainly improves reasoning-eligible accuracy, while DB mainly reduces exactly-N failures, with little additional RE accuracy gain from adding DB to CG.

5. **The method’s advantage is specific rather than universal.**
   This is supported by the null or weak results on stochastic and order-sensitive conditions, where the architecture does not show a reliable advantage.

Overall, the evidence supports the narrower architectural claim, but stronger claims about open-ended or broadly general hypothesis-space restructuring should be stated more cautiously.

**Requested Changes:**

1. **Narrow the central claim.**
   Please revise the paper to make clear that the current evidence supports pre-specified runtime monitoring and structured state-graph expansion, rather than open-ended hypothesis-space restructuring.

2. **Clarify what is engineered versus learned.**
   The dynamic behaviors are hand-designed monitors, so the paper should explicitly distinguish the contribution of the LLM agent from the contribution of the predefined architectural scaffold.

3. **Moderate claims about general causal discovery.**
   The results show improved adaptation to a particular kind of regime change in a controlled causal-learning task, but they do not yet establish broadly general causal discovery ability.

4. **Validate generality of dynamic behaviors.**
   Please add experiments or analysis showing whether the same dynamic behavior mechanism works on regime-change variants that are less directly matched to the current DB4 trigger design.

---

> ### Author Response · Authors · 2026-07-14
> **Response to Reviewer ufSt: we propose tightening claims and reframing language to address all concerns**
>
> We thank the reviewer for the accurate summary and for endorsing the paper's main claims. We agree with the recommended tightening and propose the following.
>
> **Narrow the central claim; distinguish engineered from learned; moderate general causal-discovery language (requests #1–#3).** We agree on all three and treat them as one coherent revision. We propose to state clearly in the Introduction, not only in later sections, that the current evidence supports *pre-specified runtime monitoring and structured state-graph expansion* (i.e. selection among designer-authored hypothesis-space expansions rather than open-ended or learned hypothesis-space restructuring). We will explicitly separate the contribution of the frozen LLM from that of the hand-authored scaffold (the weights are not updated; "learning" here denotes within-episode inference over observations). And we will scope the results to adaptation under a particular class of regime change in a controlled causal-learning task, rather than general causal discovery. Much of this language already exists in §6.4 and §7; the change is to foreground it.
>
> **Validate generality of the dynamic behaviors (request #4).** We take this point seriously, and we'd note that the paper already contains evidence bearing on it. Beyond DB4 (rule change), the benchmark evaluates DB2-order and DB3-stochasticity (i.e., dynamic behaviors targeting their own conditions) and reports honestly that they yield *no reliable downstream gain* (§5.5), even though the monitors fire (e.g., the stochasticity behavior fires in 77% of stochastic episodes). This mixed picture is itself a form of generality evidence: the mechanism is not a blanket benefit, and its advantage is specific to hypothesis-space restructuring. We propose to make this reading explicit in the Discussion. We would note candidly that testing entirely new regime-change *variants* less matched to DB4's trigger would require new experiments beyond the present scope; the existing boundary conditions already probe the mechanism's limits, and we will frame that specificity as a finding rather than a gap.
>
> We're grateful for the clear and actionable review.

---

### Review · Reviewer_fkzY · 2026-05-27

**Summary Of Contributions:**

This paper considers causal discovery using LLMs, focussing on 'blicket detector' settings, i.e. inferring simple compositional causal rules over actions/responses, here extended to a setting where those rules themselves adapt over time. Specifically the focus is on discovery of causes from small amounts of interventional data gathered by an LLM agent about an abstract/unfamiliar scenario, as opposed to merely answering questions about causality based on knowledge of expected structures acquired explicitly during pretraining. Firstly, the work contributes a benchmark intended to show whether LLMs can discover 'dynamic' causal rules in the case that they change mid-episode. It also conducts experiments on a specific agentic setup, claiming that a combination of two components help – use of a 'context graph' and support for 'dynamic behaviors'. These are not learnt modules (nor is the LLM itself adapted), but simplify scaffolding around a pretrained model. The context graph is a structured text representation of the current state of the agent, its discovery process, and of its current beliefs about rules of the world. Dynamic behaviors refers to adaptation to changing rules, implemented here by a pipeline that checks whether the current rules still seem to explain the observations (asking another LLM to assess this). Experiments on a dynamic extension of the 'blicket detector' setup from cognitive psychology (determining which of several objects triggers an event, and whether all/one are needed to do so) show that this scaffolding results in significantly more accurate causal discovery than the underlying LLM by itself.

**Audience:**

Yes

**Audience Explanation:**

I feel the paper is of sufficient interest to the community. Although this is not causal discovery in the most traditional sense, the work is likely of some interest to that audience, while it is also relevant to work on tool-augmented / scaffolded LLMs and agents (including their evaluation on atypical tasks).

The proposed benchmark itself (task, metrics and protocol) is likely more broadly useful than the specific architecture / scaffolding developed around (and validated solely on) Claude.

The work is less compelling as a general claim about causal reasoning in AI given its focus on Blicket-style synthetic tasks with highly specific prompting, but it still asks (and roughly answers) a worthwhile question of whether agent scaffolding can enable certain forms of adaptive reasoning where pure scaling does not yet seem to.

The paper is clear, well-organised and pleasant to read throughout.

**Broader Impact Concerns:**

This work does not raise any significant concerns. There is a short and entirely adequate broader impact statement.

**Claims And Evidence:**

Yes

**Claims Explanation:**

Overall the experiments provide fairly convincing evidence for the claims regarding the combined benefit of the 'context graph' and 'dynamic behaviors' aspects. There are detailed ablations, and the proposed benchmark setup provides very direct evidence in the relevant settings / protocols.

There are still some (relatively minor) issues with the experiments – in particular, only one model is used, meaning the generality cannot be assessed; and the 'one shot' assumption of the dynamic behaviors (making the setting very artificial). Of course the blicket-style problems themselves are a highly artificial domain, albeit one well established in cognitive/developmental psychology. There is a reasonable and balanced discussion of limitations of the experiments, and their impact on the conclusions.

On the slightly broader level, the claim about necessity and sufficiency of this particular scaffolding setup for causal discovery with LLMs seems a little strong; this could be mitigated by rewording lightly. In particular, the possibility that similar results could be achieved via other routes remains.

As a high-level point, the chosen setting of the paper is still to use a pretrained (and frozen) LLMs as a base source of knowledge, then to do causal discovery on a well-defined domain with specific text prompts etc (guiding the LLM to experiment in a certain way, report its results via the specific DSL, and to make hypotheses consistent with its collected observations). This is far from the setting that blicket-style experiments on humans are typically conducted, where there is a genuine learning process occurring, not just a scaffolding-mediated test-time inference at the 'meta-level', not affecting the weights/knowledge in the model itself.

**Requested Changes:**

As noted above, the paper is already largely acceptable, and I enjoyed reading it. Still, I would appreciate it if the claims were slightly adjusted as mentioned above under 'claims' – in particular the second and third contributions in the introduction are rather grand and general sounding as-is, and should perhaps be scoped more tightly to the specific experiments. While these two things do indeed help, the reader might be left with an impression of more general applicability / necessity / sufficiency beyond the specific setup investigated here.

The term "architecture" is used in several, places, but I feel this is a little misleading due to its (over)use specifically to mean the 'internal' structure of a neural network; here it refers to an agentic and

---

> ### Author Response · Authors · 2026-07-14
> **Response to Reviewer fkzY: we propose tightening claims and reframing language to address all concerns**
>
> We thank the reviewer for the careful summary and the positive overall assessment, and we're glad the paper read as clear and well-organized. We agree with both requested adjustments and propose the concrete changes below.
>
> **Scope of contributions 2–3.** We agree these are worded more generally than the experiments warrant. We propose to scope them explicitly to the Extended Blicket Benchmark with the tested models, and to soften the necessity/sufficiency language. Statements such as "the architecture requires both components" would become "within this architecture, each component addresses a distinct failure mode, and neither alone was sufficient in our experiments." This preserves the empirical separable-pathways finding while removing any implication of a *unique* or universally necessary route to the capability. We believe this addresses the reviewers point what the evidence can and cannot establish and other routes may well exist.
>
> **The term "architecture."** This is well taken. We use "architecture" to mean agent-level scaffolding around a frozen model, not neural-network architecture, and the title's "architectural scaffolding" reflects that intent. We propose to add a brief gloss at first use to make this unambiguous, so the term is not read in its network-internal sense.
>
> **Frozen model / test-time scaffolding.** We agree with the framing that this is scaffolding-mediated test-time inference rather than a learning process affecting the model's weights, and that this differs from the genuine learning in human blicket studies. The manuscript states this in §6.4 and §7; to address the reviewer's concern, we would foreground it earlier (Introduction/§3) so the distinction is unmistakable, rather than leaving it to Future Work and Limitations.
>
> On generality: we acknowledge the single-model limitation directly. Only two models were used (Sonnet 4.5 as the action model and Haiku 4.5 as the lightweight judge for dynamic-behavior triggers) and we make no cross-model claim. We would frame single-model as a disclosed scope boundary and note that the released benchmark, traces, and analysis scripts are designed precisely to let others test cross-model generality.
>
> We believe these are all wording-level changes consistent with the reviewer's "already largely acceptable" assessment, and we're grateful for the constructive framing.

---

### Review · Reviewer_xH1X · 2026-07-01

**Summary Of Contributions:**

This paper makes three claims:

1. **The Extended Blicket Benchmark.** The authors extend the blicket detector paradigm (Kosoy et al., 2022a) into a parameterized evaluation suite for hypothesis-space restructuring. The key addition is a *hidden moderator* condition where the causal rule changes mid-experiment (e.g., conjunctive A,B → disjunctive C), requiring the agent to detect the regime shift and re-learn. The paper also contributes *reasoning-eligible (RE) accuracy*, a metric that filters out structurally predetermined episodes (pre-switch and exactly-N) to isolate genuine reasoning capacity.

2. **The Separable Pathways Finding.** The paper evaluates a compositional architecture with two components — *context graphs* (typed state machines structuring exploration) and *dynamic behaviors* (runtime monitors that detect hypothesis-space inadequacy and expand the graph). Across 1,085 trials, these components make orthogonal contributions: context graphs drive reasoning quality within the post-switch hypothesis space (accounting for 94% of RE accuracy gain, Base 73.3% → CG 93.9%), while dynamic behaviors drive reasoning eligibility by detecting regime changes and preventing exactly-N structural traps (CG 28.0% → CG+DB 6.0%, p = 0.003).

3. **Boundary Conditions.** Two additional extended conditions — order-sensitive and stochastic — show no reliable architectural advantage, establishing that the benefit is specific to hypothesis-space restructuring and absent for statistical inference under noise or combinatorial search in information-sparse environments.

**Audience:**

Yes

**Audience Explanation:**

The paper addresses a genuine gap in the literature — the disconnect between developmental evidence that causal learning requires hypothesis revision and the empirical finding that LLMs reproduce but cannot revise causal knowledge. The separable pathways finding, where two architectural components address statistically independent failure modes, is a clean result that should interest researchers working on agent architectures, causal reasoning, and evaluation methodology. The extended blicket benchmark, particularly the hidden moderator condition and the reasoning-eligible accuracy metric, provides a reusable instrument that the community can adopt independently of the proposed architecture. The exactly-N failure mode is a broadly applicable insight for any evaluation involving hidden state changes.

The paper will be of moderate-to-high interest to researchers in: LLM agent architectures, causal reasoning evaluation, developmental-AI connections, and structured prompting. It will be of lower interest to the broader ML community due to the narrow experimental scope (one task, one model family).

**Broader Impact Concerns:**

No significant concerns. The benchmark environment is synthetic, the paper makes no deployment claims, and the broader impact statement (Section 8.1) appropriately notes that real-world applications would require human oversight and domain-specific validation. The paper's connection to clinical/regulatory analogues (treatment protocols, financial regulation) is framed as motivating the research direction, not as claiming deployment readiness.

**Claims And Evidence:**

No

**Claims Explanation:**

**Partially**
There are claims that are NOT adequately supported or are overclaimed:

**The "hypothesis-space restructuring" framing is epistemologically misleading.** The paper draws an extended analogy between dynamic behaviors and *overhypotheses* from developmental psychology (Kemp et al., 2007; Tenenbaum et al., 2011). In the developmental framework, children *induce* overhypotheses from experience — they learn that "this domain uses conjunctive rules" by observing patterns across multiple instances. Here, the overhypotheses are **pre-compiled by the designer**: the four dynamic behaviors (exploration_stagnation, order_hypothesis, stochasticity_hypothesis, rule_change_hypothesis) are hand-authored monitors with hand-authored trigger prompts, thresholds, and graph modifications. The agent *selects* among them via a lightweight Haiku evaluator, but never *constructs* new representational categories. What the paper calls "hypothesis-space restructuring" is more precisely "selection from a designer-specified menu of hypothesis-space expansions." The paper acknowledges this gap in Section 6.4 and proposes cross-episode induction as future work, but the framing throughout the main text (particularly the overhypothesis mapping in Section 3.3 and Table 2) implies a deeper analogy than the architecture actually supports. This is the most significant overclaim in the paper.

**The claim that "hypothesis-space restructuring is an architectural capability" (Section 6.1) overreaches from the evidence.** What the experiments show is that *this particular architectural scaffolding* helps *this particular LLM* on *this particular benchmark*. The paper tests only Claude Sonnet 4.5 + Haiku 4.5. Without cross-model validation (GPT-4, Gemini, open-weight models), the claim that the benefit is "architectural, not purely a function of training data or scale" cannot be distinguished from the alternative that the benefit is specific to how Claude models respond to structured prompting. The authors acknowledge this in Section 7 (Limitations), but the main text's claim language is stronger than the evidence warrants.

**The "94% of the gain" attribution is precise but potentially misleading.** The paper attributes 94% of the Base-to-CG+DB RE accuracy improvement to context graphs (CG accounts for +20.6pp out of +22.0pp total). But the CG → CG+DB comparison (93.9% → 95.3%) has only n = 33 vs n = 43 RE episodes and is not statistically significant (p = 1.000). The 94% figure therefore reflects the CG effect being large and the residual being within noise — it does not positively demonstrate that dynamic behaviors contribute nothing to reasoning quality. The paper handles this correctly in the text but the "94%" figure in the abstract and conclusion may be read as a precise causal attribution rather than a decomposition observation.

**Missing baselines weaken the architectural claims.** The only baseline is a bare LLM with a task description. There are no comparisons to chain-of-thought prompting, ReAct, tree-of-thought, or other structured prompting methods. There are no comparisons to other state-machine agent frameworks (StateFlow, AFlow). Without these baselines, it is impossible to determine whether the *specific* architectural choices (typed state graphs, dynamic behavior monitors) are necessary, or whether *any* structured prompting that guides systematic exploration would achieve similar gains.

**Requested Changes:**

### Critical (must address before acceptance):

1. **Reframe the overhypothesis analogy honestly throughout the main text.** The current framing in Sections 3.3, Table 2, and the Discussion implies that dynamic behaviors implement something analogous to overhypothesis *learning*. They do not — they implement overhypothesis *selection* from a pre-compiled set. The paper should clearly state this distinction in the introduction and maintain it consistently. The Section 6.4 acknowledgment is insufficient because most readers will form their impression from the main text. Suggested revision: replace "computational analogue to overhypotheses" with "computational analogue to overhypothesis *selection*" and explicitly note that the repertoire is fixed at design time.

2. **Add structured prompting baselines.** At minimum, compare to: (a) chain-of-thought prompting with explicit instructions to consider rule changes, and (b) a simpler structured prompt that provides the same exploration guidelines as the context graph but without the state-machine formalism. Without these, the contribution of the *specific architectural mechanism* (typed state graphs with validated transitions) versus the contribution of *any systematic exploration guidance* cannot be assessed. This is important because the context graph's guidelines include domain-relevant vocabulary ("test individual objects," "vary arrangement and sequence") that may be doing most of the work independent of the graph structure.

3. **Temper the "architectural capability" claim.** The current claim in Section 6.1 that "hypothesis-space restructuring is partially architectural, and not purely a function of training data or scale" requires cross-model evidence. Either test on at least one additional model family, or soften the claim to "the results are consistent with an architectural contribution, pending cross-model validation."

### Recommended (would strengthen the paper):

4. **Report power analyses.** The paper notes that "smaller effects... may require substantially larger samples to resolve" (Section 7) but does not report post-hoc power for any comparison. Given that several key comparisons involve n = 30–50 per cell, reporting achieved power would help readers assess what the non-significant results (particularly the CG → CG+DB RE accuracy comparison) actually tell us.

5. **Discuss the co-development risk more prominently.** The iterative development of both the benchmark and the architecture across 11+ runs (Section 7, "Iterative development") is a significant concern. The paper's mitigation arguments (large effect sizes, boundary condition null results) are reasonable but this should appear in the main text discussion, not only in the limitations section.

6. **Clarify the exactly-N classification edge case.** The paper states that exactly-N episodes have "0% accuracy by definition" because "the agent submitted before observing any post-switch evidence." This is correct when the switch triggers on the Nth *successful activation* and the agent submits immediately. But could an agent observe the switch happening (e.g., the Nth activation itself produces an anomalous result) and still submit at exactly N? If the switch occurs *on* the Nth activation, the activation itself might constitute post-switch evidence. Clarify the timing semantics.

7. **Expand the order-sensitive analysis.** The null result on the order-sensitive condition (p = 0.276) is based on only n = 30 per cell with a 4-object configuration. Given the 10pp directional effect, this may be underpowered. Either run a larger sample or discuss the power limitation explicitly.

8. **Consider the cost-benefit tradeoff.** The CG+DB agent incurs a 30–50% cost premium over Base (Section 4.4) due to Haiku-based trigger evaluations. The paper should discuss whether the accuracy gains justify this cost, especially since the base LLM achieves 73.3% RE accuracy without any scaffolding.

---

> ### Author Response · Authors · 2026-07-14
> **Response to Reviewer xH1X: we address all points with either precision or tightening claims**
>
> We thank the reviewer for a careful and constructive reading. The critique converges on a central theme, namely that several claims are pitched more broadly than the evidence licenses, and we agree with the criticism in general. We think most of these gaps can be closed by tightening claims and foregrounding qualifications already present in the manuscript. We structure our response below with where we agree, what we propose to change, and one point where we would push back respectfully.
>
> **1. The overhypothesis framing (selection vs. induction, requested #1).** We accept this as the most important revision and we agree with the distinction. Our architecture performs *selection among* a designer-specified repertoire of hypothesis-space expansions; it does not *induce* new ones from experience. The manuscript already states this (§6.4 describes our setup as testing "selection among pre-compiled overhypotheses rather than induction of new ones," with "the set of possible restructurings fixed at design time") but the reviewer is right that a reader forms their impression on overhypotheses from §3.3 and Table 2, not from Future Work at the end of the manuscript. We propose to (i) state the selection-not-induction point at first use in the Introduction, (ii) relabel the §3.3/Table 2 mapping as "computational analogue to overhypothesis *selection*," and (iii) note explicitly that the induction step in Kemp et al. (2007)/Tenenbaum et al. (2011) is not what we implement. We would retain the four-part mapping as a stated *design rationale*, since it motivated the architecture, but we will caption it as such rather than as an equivalence.
>
> **2. "Architectural capability" overreaches without cross-model evidence (requested #3).** Agreed, and we would soften accordingly. We propose to replace "the gap … is partially architectural, and not purely a function of training data or scale" with a claim the single-model evidence supports: that our results are *consistent with* an architectural contribution, and that distinguishing this from a model-specific effect requires cross-model validation (already flagged in §7). The Introduction's contributions 2–3 would be scoped to the Extended Blicket Benchmark with the tested models.
>
> **3. The "94%" attribution.** We take the point that the figure can read as a precise causal attribution rather than a decomposition, and we propose to qualify it wherever it appears. We can now state the decomposition concretely: the Base→CG gain of +20.6pp has a 95% CI of [+2.2, +38.9] (excludes zero), whereas the CG→CG+DB residual of +1.4pp has a CI of [−10.3, +15.4] (straddles zero). So "≈94%" describes how the gain decomposes, not a demonstration that dynamic behaviors contribute nothing to reasoning quality. We will make this distinction explicit in the abstract and conclusion.
>
> **4. Making clear what the CG→CG_DB null tells readers (requested #4)** We agree this should be stated explicitly, and we propose to add a short passage doing so. The key point is that this comparison cannot settle whether dynamic behaviors add anything to reasoning quality. This is not because we found nothing, but because there is almost no room left to measure. CG already reaches 93.9%, so the most dynamic behaviors could add is about six percentage points. But at these cell sizes we could only reliably detect a difference larger than roughly 23. Any real effect is therefore too small for this experiment to catch, so the null should be read as uninformative about a small remaining effect, not as evidence of none.
>
> **5. Exactly-N timing, co-development risk, cost–benefit (requested #5, #6, #8).** These we can address all of these directly: clarifying that the rule switches *after* the N-th activation (so the first post-switch evidence appears only on the next test); lifting a short form of the iterative-development caveat from §7 into the main Discussion; and adding a cost–benefclait sentence around the reported 30–50% Haiku premium.
>
> **Where we would respectfully push back:**
>
> *Structured-prompting baselines (requested #2, critical).* We agree the underlying point is real, namely that the context graph bundles typed structure with domain-relevant guideline vocabulary, and we cannot separate formalism from good instructions. We propose to concede this explicitly, scoping every context-graph claim to the graph as a whole and flagging the structure-vs-content decomposition as future work. Our belief is that the paper's question is whether these two components make separable contributions, which the additive Base→CG→CG+DB hierarchy isolates. So, a horse-race against ReAct/StateFlow/AFlow answers a different question. We note also that the Base agent is already prompted for explicit step-by-step reasoning, so a chain-of-thought comparator is in effect present. Once claims are scoped as above, we believe the contribution stands without these additional systems.
>
> We hope this addresses the reviewer's core concerns.

---

### Author Response · Authors · 2026-07-14
**Response to the Action Editor: we propose tightening claims and adding more precise language to address all concerns**

We thank the three reviewers for constructive and largely convergent feedback. Two reviewers (fkzY, ufSt) assessed the claims as supported and the work of interest to TMLR's audience; the third (xH1X) shares this positive view of the contribution's interest and raises a coherent set of concerns about the scope of our claims, some of which overlap with the other reviews. We read the reviews as agreeing on a single actionable theme, and we summarize our planned response here; per-reviewer replies give more detail.

**The common theme is scope, and it is addressable by tightening claims, not by new experiments.** Across all three reviews the recurring request is to scope the central claims to the specific benchmark, model, and experimental setting, and to distinguish our designer-authored, test-time scaffolding from learned or open-ended hypothesis-space restructuring. We agree, and we propose a consistent set of wording revisions: (i) reframing the overhypothesis analogy as overhypothesis *selection* from a fixed repertoire; (ii) softening "architectural capability … not scale" to a claim consistent with our single-model evidence, pending cross-model validation; (iii) foregrounding in the Introduction (rather than only in §6.4/§7) that the LLM is frozen and the scaffolding operates at test time; (iv) softening necessity/sufficiency language; and (v) glossing "architecture" as agent-level scaffolding. Notably, much of this qualifying language is already present in the manuscript; the revision foregrounds it so a reader's impression is formed from the early on in main text.

**We would also add targeted precision** where reviewers asked for it: a principled treatment of the CG→CG+DB null (design sensitivity), a qualifier making the "94%" figure read as a decomposition (with supporting confidence intervals), clarified exactly-N timing semantics, the co-development caveat surfaced into the main Discussion, and a cost–benefit note. None of these requires new data; all existing manuscript statistics have been re-verified.

**Two requests we propose to address by scoping rather than by new experiments.** First, additional structured-prompting and agent-framework baselines (xH1X): we concede the fair core of this point, namely that the context graph bundles structure with guideline content. We will scope our claims to the context graph as a whole while flagging the structure-vs-content decomposition as future work. However, the paper's central question is the *separability* of two components, which the additive ablation isolates, rather than a comparison against alternative frameworks. Second, on cross-model replication (only two models were used, Sonnet 4.5 as action model, Haiku 4.5 as trigger judge), we make no cross-model claim and we would present the single-model setup as a disclosed scope boundary, with released artifacts enabling others to extend it. We believe these scoping choices bring the claims into line with the evidence, which we understand to be the operative acceptance criterion.

We believe that the resulting revisions are minor in scope and responsive to the substance of all three reviews. We would be glad to adjust further if the reviewers feel any specific claim remains broader than the evidence supports.